# GENERATIVE HUMAN MOTION STYLIZATION IN LATENT SPACE

**Chuan Guo[1*], Yuxuan Mu[1*], Xinxin Zuo[2], Peng Dai[2], Youliang Yan[2], Juwei Lu[2], Li Cheng[1]**
[1]University of Alberta,    [2]Noah's Ark Lab, Huawei Canada

## ABSTRACT

Human motion stylization aims to revise the style of an input motion while keeping its content unaltered. Unlike existing works that operate directly in pose space, we leverage the *latent space* of pretrained autoencoders as a more expressive and robust representation for motion extraction and infusion. Building upon this, we present a novel *generative* model that produces diverse stylization results of a single motion (latent) code. During training, a motion code is decomposed into two coding components: a deterministic content code, and a probabilistic style code adhering to a prior distribution; then a generator massages the random combination of content and style codes to reconstruct the corresponding motion codes. Our approach is versatile, allowing the learning of probabilistic style space from either style labeled or unlabeled motions, providing notable flexibility in stylization as well. In inference, users can opt to stylize a motion using style cues from a reference motion or a label. Even in the absence of explicit style input, our model facilitates novel re-stylization by sampling from the unconditional style prior distribution. Experimental results show that our proposed stylization models, despite their lightweight design, outperform the state-of-the-arts in style reenactment, content preservation, and generalization across various applications and settings.

## 1 INTRODUCTION

The motions of our humans are very expressive and contain a rich source of information. For example, by watching a short duration of one individual's walking movement, we could quickly recognize the person, or discern the mood, age, or occupation of the person. These distinguishable motion traits, usually thought of as styles, are therefore essential in film or game industry for realistic character animation. It is unfortunately unrealistic to acquire real-world human motions of various styles solely by motion capture. Stylizing existing motions using a reference style motion (*i.e.,* motion-based), or a preset style label (*i.e.,* label-based) thus becomes a feasible solution.

Deep learning models have recently enabled numerous data-driven methods for human motion stylization. These approaches, however, still find their shortfalls. A long line of existing works (Aberman et al., 2020; Holden et al., 2016; Jang et al., 2022; Tao et al., 2022) are limited to deterministic stylization outcomes. (Park et al., 2021; Wen et al., 2021) though allows diverse stylization, their results are far from being satisfactory, and the trained models struggle to generalize to other motion datasets. Furthermore, all of these approaches directly manipulate style within raw poses, a redundant and potentially noisy representation of motions. Meanwhile, they often possess rigid designs, allowing for only supervised or unsupervised training, with style input typically limited to either reference motions or labels, as shown in Tab. 1.

In this work, we introduce a novel *generative* stylization framework for 3D human motions. Inspired by the recent success of content synthesis in latent space (Guo et al., 2022a; Chen et al., 2022; Rombach et al., 2022; Ramesh et al., 2022), we propose to use *latent* motion features (namely motion code) of pretrained convolutional autoencoders as the intermedia for motion style extraction and infusion. Compared to raw poses, the benefits are three-folds: **(i)** Motion codes are more compact and expressive, containing the most discriminative features of raw motions; **(ii)** Autoencoders can be

---

*These authors contributed equally to this work.

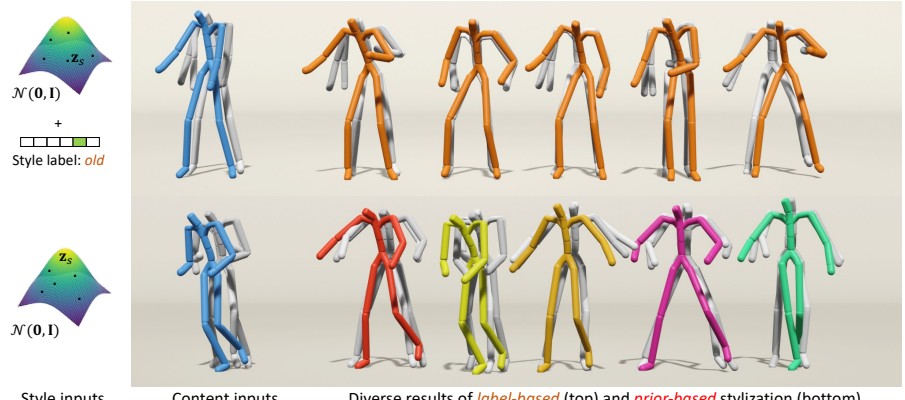

Style inputs | Content inputs | Diverse results of *label-based* (top) and *prior-based* stylization (bottom)

Figure 1: (**Top**) Given an input motion and target style label (*i.e., old*), our label-based stylization generates diverse results following provided label. (**Bottom**) Without any style indicators, our prior-based method randomly re-stylizes the input motion using sampled prior styles $\mathbf{z}_s$. Five distinct stylized motions from the same content are presented, with poses synchronized and history in gray. See Fig. 3 (b) and (d) for implementations.

learned once on a large dataset and reused for downstream datasets. Thanks to the inductive bias of CNN (LeCun et al., 1995), the learned motion code features typically contains less noise, resulting in improved generalization, as empirically demonstrated in Tables 2 and 11; **(iii)** Practically, motion code sequences are much shorter than motions, making them more manageable in neural networks. Building on this, our latent stylization framework decomposes the motion code into two components: a temporal and deterministic *content* code, and a global probabilistic *style* code confined by a prior Gaussian distribution. The subsequent generator recombines content and style to synthesize valid motion code. During training, besides auto-encoding and decoding, we swap the contents and styles between random pairs, and the resulting motion codes are enforced to recover the source contents and styles through cycle reconstruction. To further improve content-style disentanglement, we propose a technique called *homo-style alignment*, which encourages the alignment of style spaces formed by different motion sub-clips from the same sequence. Lastly, the global velocity of resulting motions are obtained through a pre-trained global motion regressor.

Our approach offers versatile stylization capabilities (Tab. 1), accommodating various conditioning options during both training and inference: 1) Deterministic stylization using style from **exemplar motions**; 2) In the label conditioned setting, our model can perform diverse stylization based on provided **style labels**, as in Fig. 1 (top); 3) In the unconditional setting, our model can randomly sample styles from the **prior distribution** to achieve stochastic stylization, as in Fig. 1 (bottom). Benefiting from our latent stylization and lightweight model design, our approach achieves state-of-the-art performance while being 14 times faster than the most advanced prior work (Jang et al., 2022), as shown in Table 5.

Our key contributions can be summarized as follows. Firstly, we propose a novel generative framework, using motion latent features as an advanced alternative representation, accommodating various training and inference schemes in a single framework. Secondly, through a comprehensive suite of evaluations on three benchmarks, our framework demonstrates robust and superior performance across all training and inference settings, with notable efficiency gains.

## 2 RELATED WORK

**Image Style Transfer.** Image style in computer vision and graphics is typically formulated as the global statistic features of images. Early work (Gatys et al., 2016) finds it possible to transfer the visual style from one image to another through aligning their Gram matrices in neural networks. On top of this, (Johnson et al., 2016; Ulyanov et al., 2016a) enable faster transferring through an additional feed-forward neural networks. The work of (Ulyanov et al., 2016b) realizes that instance normalization (IN) layer could lead to better performance. However, these works can only be applied on single style image. (Huang & Belongie, 2017) facilitates arbitrary image style transfer by introducing adaptive instance normalization (AdaIN). Alternatively, in PatchGAN (Isola et al., 2017) and CycleGAN (Zhu et al., 2017), textures and styles are translated between images by ensuring the local similarity using patch discriminator. Similar idea was adopted in (Park et al., 2020),

| | Supervised (*w* style label) | | Unsupervised (*w/o* style label) | | Generative |
|---|---|---|---|---|---|
| | Motion-based | Label-based | Motion-based | Prior-based | |
| (Xia et al., 2015) | | ✓ | | | |
| (Holden et al., 2016; 2017) | | | ✓ | | |
| (Aberman et al., 2020) | ✓ | | | | |
| (Park et al., 2021) | ✓ | ✓ | | | ✓ |
| (Tao et al., 2022) | ✓ | | | | |
| (Jang et al., 2022) | | | ✓ | | |
| Ours | ✓ | ✓ | ✓ | ✓ | ✓ |

Table 1: Our generative framework owns flexible design for training and inference.

which proposes patch co-occurrence discriminator that hypothesizes images with similar marginal and joint feature statistics appear perceptually similar.

**Motion Style Transfer.** Motion style transfer has been a long-standing challenge in computer animation. Early work (Xia et al., 2015) design an online style transfer system based on KNN search. (Holden et al., 2016; Du et al., 2019; Yumer & Mitra, 2016) transfers the style from reference to source motion through optimizing style statistic features, such as Gram matrix, which are computationally intensive. Feed-forward based approaches (Holden et al., 2017; Aberman et al., 2020; Smith et al., 2019) properly address this problem, where (Aberman et al., 2020) finalizes a two-branch pipeline based on deterministic autoencoders and AdaIN (Huang & Belongie, 2017) for style-content disentanglement and composition; while (Smith et al., 2019) manages to stylize existing motions using one-hot style label, and models it as an class conditioned generation process. More recently, with the explosion of deep learning techniques, some works adopt graph neural networks (GNN) (Park et al., 2021; Jang et al., 2022), advanced time-series model (Tao et al., 2022; Wen et al., 2021), or diffusion model (Raab et al., 2023) to the motion style transfer task. Specifically, (Jang et al., 2022) realizes a framework that extracts style features from motion body parts.

**Synthesis in Latent.** Deep latent have been evidenced as a promising alternative representation for content synthesis including images (Rombach et al., 2022; Ramesh et al., 2022; Esser et al., 2021b;a; Ding et al., 2021), motion (Guo et al., 2022a;b; Gong et al., 2023; Chen et al., 2022), 3D shape (Zeng et al., 2022; Fu et al., 2022), and video (Yan et al., 2021; Hu et al., 2023). These works commonly adopt a two-stage synthesis strategy. At the first stage, the source contents are encoded into continuous latent maps (*e.g.*, using autoencoders, CLIP (Radford et al., 2021)), or discrete latent tokens through VQ-VAE (Van Den Oord et al., 2017). Then, models are learned to generate these latent representation explicitly which can be inverted to data space in the end. This strategy has shown several merits. Deep latent consists of the most representative features of raw data, which leads to a more expressive and compact representation. It also cuts down the cost of time and computation during training and inference. These prior arts inspire the proposed latent stylization in our approach.

## 3 GENERATIVE MOTION STYLIZATION

An overview of our method is described in Figure 2. Motions are first projected into the latent space (Sec. 3.1). With this, the latent stylization framework learns to extract the content and style from the input code (Sec. 3.2), which further support multiple applications during inference (Sec. 3.3).

### 3.1 MOTION LATENT REPRESENTATION

As a pre-processing step, we learn a motion autoencoder that builds the mapping between motion and latent space. More precisely, given a pose sequence $\mathbf{P} \in \mathbb{R}^{T \times D}$, where $T$ denotes the number of poses and $D$ pose dimension, the encoder $\mathcal{E}$ encodes $\mathbf{P}$ into a motion code $\mathbf{z} = \mathcal{E}(\mathbf{P}) \in \mathbb{R}^{T_z \times D_z}$, with $T_z$ and $D_z$ the temporal length and spatial dimension respectively, and then the decoder $\mathcal{D}$ recovers the input motion from the latent features, formally $\hat{\mathbf{P}} = \mathcal{D}(\mathbf{z}) = \mathcal{D}(\mathcal{E}(\mathbf{P}))$.

A well-learned latent space should exhibit smoothness and low variance. In this work, we experiment with two kinds of regularization methods in latent space: 1) as in VAE (Kingma & Welling, 2013), the latent space is formed under a light KL regularization towards standard normal distribution $\mathcal{L}_{kld}^{l} = \lambda_{kld}^{l} D_{\mathrm{KL}}(\mathbf{z} || \mathcal{N}(\mathbf{0}, \mathbf{I}))$ ; and 2) similar to (Guo et al., 2022a), we train the classical autoencoder and impose L1 penalty on the magnitude and smoothness of motion code sequences,

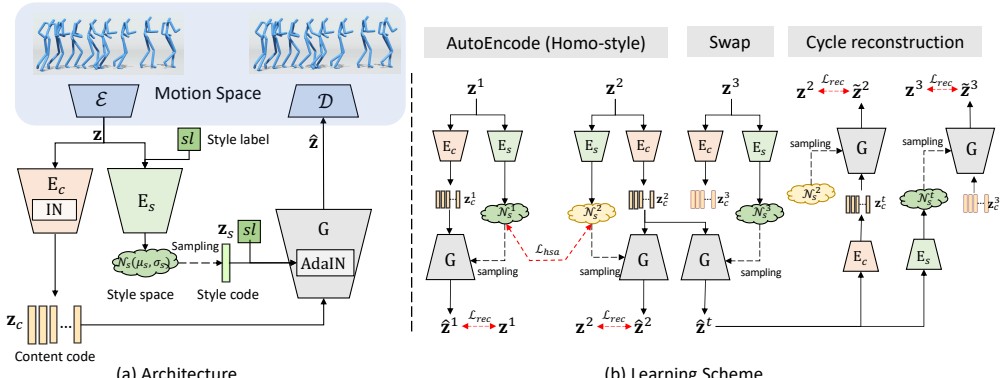

Figure 2: **Approach overview.** (a) A pre-trained autoencoder $\mathcal{E}$ and $\mathcal{D}$ (Sec. 3.1) builds the mappings between *motion* and *latent* spaces. Motion (latent) code $\mathbf{z}$ is further encoded into two parts: content code $\mathbf{z}_c$ from content encoder ($E_c$), and style space $\mathcal{N}_s$ from style encoder ($E_s$) that take style label $sl$ as an additional input. The content code ($\mathbf{z}_c$) is decoded back to motion code ($\hat{\mathbf{z}}$) via generator G. Meanwhile, a style code $\mathbf{z}_s$ is sampled from style space ($\mathcal{N}_s$), together with style label ($sl$), which are subsequently injected to generator layers through adaptive instance normalization (AdaIN). (b) Learning scheme, where style label ($sl$) is omitted for simplicity. Our model is trained by autoencoding for content and style coming from the **same** input. When decoding with content from **different** input (*i.e.,* swap), we enforce the resulting motion code ($\hat{\mathbf{z}}^t$) to follow the cycle reconstruction constraint. For motion codes ($\mathbf{z}^1$, $\mathbf{z}^2$) segmented from the same sequence (homo-style), their style spaces are assumed to be close and learned with style alignment loss $\mathcal{L}_{hsa}$.

giving $\mathcal{L}^l_{reg} = \lambda_{l1}\|\mathbf{z}\|_1 + \lambda_{sms}\|\mathbf{z}_{1:T_z} - \mathbf{z}_{0:T_z-1}\|_1$. Our motion encoder $\mathcal{E}$ and decoder $\mathcal{D}$ are simply 1-D convolution layers with downsampling and upsampling scale of 4 (*i.e.,* $T = 4T_z$), resulting in a more compact form of data that captures temporal semantic information.

## 3.2 Motion Latent Stylization Framework

As depicted in Figure 2, our latent stylization framework aims to yield a valid parametric style space, and meanwhile, preserve semantic information in content codes as much as possible. This is achieved by our specific model design and dedicated learning strategies.

### 3.2.1 Model Architecture.

There are three principal components in our framework: a style encoder $E_s$, a content encoder $E_c$ and a generator G, as in Figure 2 (a).

**Probabilistic Style Space.** For style, existing works (Park et al., 2021; Aberman et al., 2020; Jang et al., 2022) generate deterministic style code from motion input. In contrast, our style encoder $E_s$, taking $\mathbf{z}$ and style label $sl$ as input, produces a vector Gaussian distribution $\mathcal{N}_s(\mu_s, \sigma_s)$ to formulate the style space, from which a style code $\mathbf{z}_s \in \mathbb{R}^{D^s_z}$ is sampled. In test-time, this probabilistic style space enables us to generate diverse and novel style samples.

Comparing to style features, content features exhibit more locality and determinism. Therefore, we model them deterministically to preserve the precise structure and meaning of the motion sequence. The content encoder converts the a motion code $\mathbf{z} \in \mathbb{R}^{T_z \times D_z}$ into a content code $\mathbf{z}_c \in \mathbb{R}^{T^c_z \times D^c_z}$ that keeps a temporal dimension $T^c_z$, where global statistic features (style) are erased through instance normalization (IN). The asymmetric shape of content code $\mathbf{z}_c$ and style code $\mathbf{z}_s$ are designed of purpose. We expect the former to capture local semantics while the latter encodes global features, as what style is commonly thought of. Content code is subsequently fed into the convolution-based generator G, where the mean and variance of each layer output are modified by an affine transformation of style information (*i.e.,* style code and label), known as adaptive instance normalization (AdaIN). The generator aims to transform valid combinations of content and style into meaningful motion codes in the latent space.

### 3.2.2 Learning Scheme

With the model mentioned above, we propose a series of strategies for learning disentangled content and style representations. Figure 2 (b) illustrates our learning scheme. Note the input of style label $sl$ is omitted for simplicity. During training, for each iteration, we design three groups of inputs: $\mathbf{z}^1$,

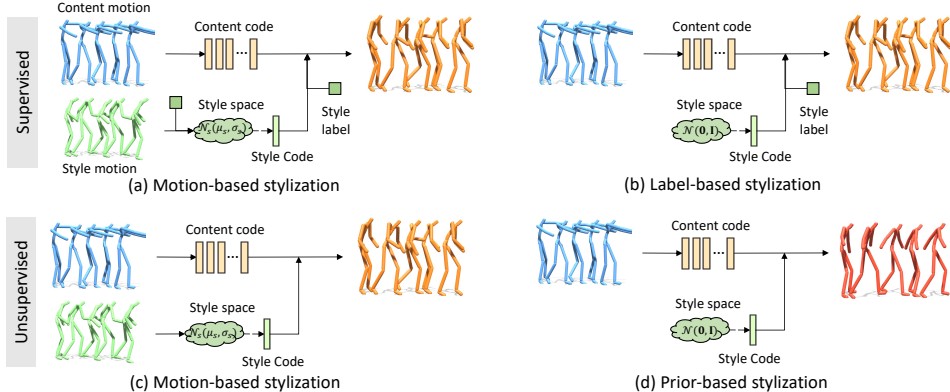

Figure 3: During inference, our approach can stylize input content motions with the style cues from (a, c) motion, (b) style label and (d) unconditional style prior space.

$\mathbf{z}^2$ and $\mathbf{z}^3$, where $\mathbf{z}^1$ and $\mathbf{z}^2$ are motion code segments coming from the same sequence and $\mathbf{z}^3$ can be any other segments.

**AutoEncoding $\mathcal{L}_{rec}$.** We train our latent stylization framework partly through autoencoding, that given motion latent codes, like $\mathbf{z}^1$ and $\mathbf{z}^2$, the generator learns to reconstruct the input from the corresponding encoded content and style features, formally $\hat{\mathbf{z}} = \mathrm{G}(\mathrm{E}_c(\mathbf{z}), \mathrm{E}_s(\mathbf{z}))$. For accurate reconstruction, we decode the resulting motion latent codes ($\hat{\mathbf{z}}^1$ and $\hat{\mathbf{z}}^2$) back to motion space ($\hat{\mathbf{P}}^1$ and $\hat{\mathbf{P}}^2$) through $\mathcal{D}$, and apply L1-distance reconstruction in both latent and motion space:

$$\mathcal{L}_{rec} = \sum_{i \in \{1,2\}} \|\hat{\mathbf{z}}^i - \mathbf{z}^i\|_1 + \|\hat{\mathbf{P}}^i - \mathbf{P}^i\|_1 \tag{1}$$

**Homo-style Alignment $\mathcal{L}_{hsa}$.** For the motion segments in one motion sequence, we could usually assume their styles are similar in all aspects. This is a strong supervision signal especially when style annotation is unavailable, dubbed *homo-style alignment* in our work. Since $\mathbf{z}^1$ and $\mathbf{z}^2$ belong to the same sequence, their learned style spaces are enforced to be close:

$$\mathcal{L}_{hsa} = D_{\mathrm{KL}}(\mathcal{N}_s^1(\mu_s^1, \sigma_s^1) \| \mathcal{N}_s^2(\mu_s^2, \sigma_s^2)) \tag{2}$$

**Swap and Cycle Reconstruction $\mathcal{L}_{cyc}$.** To further encourage content-style disentanglement, we adopt a cycle consistency constraint (Zhu et al., 2017; Jang et al., 2022) when content and style are swapped between different motion codes, such as $\mathbf{z}^2$ and $\mathbf{z}^3$ in Fig. 2. Specifically, the generator G takes as input the content from $\mathbf{z}^2$ and the style from $\mathbf{z}^3$, and then produces a new *transferred* motion code $\mathbf{z}^t$, which are supposed to preserve the content information from $\mathbf{z}^2$ and the style from $\mathbf{z}^3$. Therefore, if we re-combine $\mathbf{z}^t$'s content and $\mathbf{z}^2$'s style, the generator should be able to restore $\mathbf{z}^2$. The same to $\tilde{\mathbf{z}}^3$ that are recovered from the mix of $\mathbf{z}^t$'s style and $\mathbf{z}^3$'s content :

$$\mathcal{L}_{cyc} = \sum_{i \in \{2,3\}} \|\tilde{\mathbf{z}}^i - \mathbf{z}^i\|_1 + \|\tilde{\mathbf{P}}^i - \mathbf{P}^i\|_1 \tag{3}$$

To ensure smooth and samplable style spaces, we apply a KL loss regularization to all style spaces:

$$\mathcal{L}_{kl} = \sum_{i \in \{1,2,3,t\}} D_{\mathrm{KL}}(\mathcal{N}_s^i(\mu_s^i, \sigma_s^i)) \| \mathcal{N}(\mathbf{0}, \mathbf{I})) \tag{4}$$

Overall, our final objective is $\mathcal{L} = \mathcal{L}_{rec} + \lambda_{hsa}\mathcal{L}_{hsa} + \lambda_{cyc}\mathcal{L}_{cyc} + \lambda_{kl}\mathcal{L}_{kl}$. We also have experimented adversarial loss for autoencoding and cycle reconstruction as in (Park et al., 2021; Aberman et al., 2020; Tao et al., 2022), which however appears to be extremely unstable in training.

**Unsupervised Scheme (*w/o* Style Label).** Collecting style labeled motions is resource-consuming. Our approach can simply fit in the unsupervised setting with just one-line change of code during training—to drop out style label $sl$ input.

**Difference of $\mathcal{N}_s$ Learned *w* and *w/o* Style Label.** While learning with style label, since both the style encoder $\mathrm{E}_s$ and generator G are conditioned on style label, the style space is encouraged to learn style variables other than style label as illustrated in Fig. 8 (d). Whereas in the unsupervised setting where the networks are agnostic to style label, in order to precisely reconstruct motions, the style space is expected to cover the *holistic* style information, including style label (see Fig. 8 (c)).

### 3.2.3 GLOBAL MOTION PREDICTION

Global motion (*i.e.,* root velocity) is perceptually a more sensitive element than local joint motion (e.g., foot skating). However, given one motion, transferring its global motion to another style domain is challenging without supervision of paired data. Previous works commonly calculate the target global motion directly from the content motion, or enforce them to be close in training. This may fail when the transferred motion differs a lot from the source content. In our work, we propose a simple yet effective alternative, which is a small 1D convolutional network that predicts the global motion from local joint motion, simply trained on unlabeled data using objective of mean absolute error. During inference, the global motion of output can be accurately inferred from its local motion.

### 3.3 INFERENCE PHASE

As displayed in Figure 3, our approach at run time can be used in multiple ways. In *supervised* setting: a) **motion-based** stylization requires the user to provide a style motion and a style label as the style references; and b) **label-based** stylization only asks for a target style label for stylization. With sampled style codes from a standard normal distribution $\mathcal{N}(\mathbf{0}, \mathbf{I})$, we are able to stylize source content motion non-deterministically. In the case of *unsupervised* setting: c) motion-based stylization, which similarly, yields a style code from a reference motion; and d) **prior-based** stylization that samples random style codes from the prior distribution $\mathcal{N}(\mathbf{0}, \mathbf{I})$. Since there is no other pretext style indications, the output motion could carry any style trait in the style space.

## 4 EXPERIMENTS

We adopt three datasets for comprehensive evaluation. (Aberman et al., 2020) is a widely used motion style dataset, which contains 16 distinct style labels including *angry*, *happy*, *Old*, etc, with total duration of 193 minute. (Xia et al., 2015) is much smaller motion style collection (25 mins) that is captured in 8 styles, with accurate action type annotation (8 actions). The motions are typically shorter than 3s. The other one is CMU Mocap (CMU), an unlabeled dataset with high diversity and quantity of motion data. All motion data is retargeted to the same 21-joint skeleton structure, with a 10% held-out subset for evaluation. Our autoencoders and global motion regressor are trained on the union of all training sets, while the latent stylization models are trained **excursively** on (Aberman et al., 2020), using the other two for zero-shot evaluation. During evaluation, we use the styles from (Aberman et al., 2020) test sets to stylize the motions from one of the three test sets. Style space is learned based on motions of 160 poses (5.3s). Note our models supports stylization of arbitrary-length content motions. See Appendix D for implementation details and full model architectures.

**Metrics** in previous motion stylization works heavily rely on a sparse set of measurements, typically human evaluation and style accuracy. Here, we design a suite of metrics to comprehensively evaluate our approach. We firstly pre-train a style classifier on (Aberman et al., 2020) train set, and use it as a style feature extractor to compute *style recognition accuracy* and *style FID*. For dataset with available action annotation ( (Xia et al., 2015)), an action classifier is learned to extract content features and calculate *content recognition accuracy* and *content FID*. We further evaluate the content preservation using *geodesic distance* of the local joint rotations between input content motion and generated motion. *Diversity* in (Lee et al., 2019) is also employed to quantify the stochasticity in the stylization results. Further explanations are provided in Appendix E.

**Baselines.** We compare our method to three state-of-the-art methods (Aberman et al., 2020; Jang et al., 2022; Park et al., 2021) in their respective settings. Among these, (Aberman et al., 2020) and (Park et al., 2021) are supervised methods learned within GAN framework. (Park et al., 2021) learns per-label style space, and a mapping between Gaussian space and style space. At run time, it supports both deterministic motion-based and diverse label-based motion stylization.

### 4.1 QUANTITATIVE RESULTS

Table 2 and Table 3 present the quantitative evaluation results on the test sets of (Aberman et al., 2020), CMU Mocap (CMU) and (Xia et al., 2015). Note the latter two datasets are completely unseen to our latent stylization models. We generate results using motions in these three test sets

| Setting | Methods | (Aberman et al., 2020) | | | | CMU Mocap (CMU) | | | |
|---|---|---|---|---|---|---|---|---|---|
| | | Style Accuracy↑ | Style FID↓ | Geo Dis↓ | Div↑ | Style Accuracy↑ | Style FID↓ | Geo Dis↓ | Div↑ |
| | Real Motions | $0.997^{\pm002}$ | $0.002^{\pm000}$ | - | - | $0.997^{\pm002}$ | $0.002^{\pm000}$ | - | - |
| Motion-based (S) | (Aberman et al., 2020) | $0.547^{\pm016}$ | $0.379^{\pm018}$ | $0.804^{\pm003}$ | - | $0.445^{\pm009}$ | $0.508^{\pm011}$ | $0.910^{\pm002}$ | - |
| | (Park et al., 2021) | $0.891^{\pm007}$ | $0.038^{\pm003}$ | $0.531^{\pm001}$ | - | $0.674^{\pm014}$ | $0.136^{\pm011}$ | $0.663^{\pm003}$ | - |
| | Ours w/o latent | $0.932^{\pm008}$ | $0.022^{\pm002}$ | $0.463^{\pm003}$ | - | $0.879^{\pm008}$ | $0.046^{\pm004}$ | $0.636^{\pm004}$ | - |
| | Ours (V) | $\underline{0.935}^{\pm007}$ | $\mathbf{0.020}^{\pm002}$ | $0.426^{\pm003}$ | - | $\underline{0.918}^{\pm010}$ | $\mathbf{0.028}^{\pm003}$ | $\underline{0.629}^{\pm002}$ | - |
| | Ours (A) | $\mathbf{0.945}^{\pm007}$ | $\mathbf{0.020}^{\pm002}$ | $\mathbf{0.344}^{\pm002}$ | - | $\mathbf{0.918}^{\pm007}$ | $\underline{0.031}^{\pm003}$ | $\mathbf{0.569}^{\pm002}$ | - |
| Label-based (S) | (Park et al., 2021) | $\mathbf{0.971}^{\pm006}$ | $\mathbf{0.013}^{\pm001}$ | $0.571^{\pm002}$ | $0.146^{\pm009}$ | $0.813^{\pm010}$ | $0.065^{\pm007}$ | $0.693^{\pm004}$ | $0.229^{\pm019}$ |
| | Ours w/o latent | $0.933^{\pm009}$ | $0.023^{\pm002}$ | $0.447^{\pm002}$ | $\mathbf{0.174}^{\pm017}$ | $0.882^{\pm008}$ | $0.053^{\pm003}$ | $\underline{0.611}^{\pm003}$ | $\mathbf{0.266}^{\pm021}$ |
| | Ours (V) | $\underline{0.946}^{\pm007}$ | $0.020^{\pm002}$ | $\underline{0.427}^{\pm003}$ | $\underline{0.134}^{\pm016}$ | $\mathbf{0.923}^{\pm007}$ | $\mathbf{0.027}^{\pm003}$ | $0.614^{\pm002}$ | $\underline{0.193}^{\pm013}$ |
| | Ours (A) | $0.942^{\pm006}$ | $\underline{0.019}^{\pm001}$ | $\mathbf{0.344}^{\pm003}$ | $0.050^{\pm006}$ | $\underline{0.915}^{\pm005}$ | $\underline{0.031}^{\pm003}$ | $\mathbf{0.571}^{\pm003}$ | $0.067^{\pm005}$ |
| Motion-based (U) | (Jang et al., 2022) | $\underline{0.833}^{\pm010}$ | $0.047^{\pm004}$ | $0.559^{\pm003}$ | - | $0.793^{\pm009}$ | $0.058^{\pm004}$ | $0.725^{\pm004}$ | - |
| | Ours w/o latent | $0.780^{\pm014}$ | $0.048^{\pm004}$ | $\underline{0.466}^{\pm004}$ | - | $0.761^{\pm009}$ | $0.082^{\pm005}$ | $\mathbf{0.645}^{\pm003}$ | - |
| | Ours (V) | $\mathbf{0.840}^{\pm010}$ | $\mathbf{0.036}^{\pm003}$ | $0.478^{\pm004}$ | - | $\mathbf{0.828}^{\pm010}$ | $\mathbf{0.052}^{\pm004}$ | $0.672^{\pm003}$ | - |
| | Ours (A) | $0.804^{\pm011}$ | $\underline{0.040}^{\pm003}$ | $\mathbf{0.441}^{\pm003}$ | - | $\underline{0.799}^{\pm009}$ | $\underline{0.056}^{\pm003}$ | $\underline{0.648}^{\pm004}$ | - |
| Prior-based (U) | Ours w/o latent | - | - | $0.431^{\pm003}$ | $1.169^{\pm030}$ | - | - | $0.626^{\pm001}$ | $1.252^{\pm029}$ |
| | Ours (V) | - | - | $\mathbf{0.418}^{\pm003}$ | $1.069^{\pm028}$ | - | - | $\mathbf{0.611}^{\pm003}$ | $0.857^{\pm024}$ |
| | Ours (A) | - | - | $0.436^{\pm004}$ | $\mathbf{1.187}^{\pm029}$ | - | - | $0.641^{\pm002}$ | $\underline{0.949}^{\pm022}$ |

Table 2: Quantitative results on the (Aberman et al., 2020) and CMU Mocap test sets. ± indicates 95% confidence interval. **Bold** face indicates the best result, while underscore refers to the second best. (S) and (U) denote *supervised* and *unsupervised* setting. (V) VAE and (A) AE represent different latent models in Sec. 3.1.

| Setting | Methods | (Xia et al., 2015) | | | | |
|---|---|---|---|---|---|---|
| | | Style Acc↑ | Content Acc↑ | Content FID↓ | Geo Dis↓ | Div↑ |
| M-based (S) | (Aberman et al., 2020) | $0.364^{\pm011}$ | $0.318^{\pm008}$ | $0.705^{\pm014}$ | $0.931^{\pm003}$ | - |
| | (Park et al., 2021) | $0.527^{\pm006}$ | $0.441^{\pm009}$ | $0.381^{\pm010}$ | $\underline{0.698}^{\pm001}$ | - |
| | Ours w/o latent | $0.851^{\pm012}$ | $\underline{0.654}^{\pm012}$ | $0.258^{\pm007}$ | $0.707^{\pm004}$ | - |
| | Ours (V) | $\mathbf{0.934}^{\pm006}$ | $0.579^{\pm016}$ | $\underline{0.210}^{\pm004}$ | $0.716^{\pm003}$ | - |
| | Ours (A) | $\underline{0.926}^{\pm008}$ | $\mathbf{0.674}^{\pm011}$ | $\mathbf{0.189}^{\pm005}$ | $\mathbf{0.680}^{\pm003}$ | - |
| L-based (S) | (Park et al., 2021) | $0.796^{\pm007}$ | $0.311^{\pm009}$ | $0.507^{\pm011}$ | $0.770^{\pm003}$ | $0.175^{\pm014}$ |
| | Ours w/o latent | $0.843^{\pm012}$ | $\underline{0.655}^{\pm013}$ | $0.264^{\pm008}$ | $\underline{0.691}^{\pm003}$ | $\mathbf{0.281}^{\pm032}$ |
| | Ours (V) | $\mathbf{0.944}^{\pm008}$ | $0.606^{\pm013}$ | $\underline{0.208}^{\pm005}$ | $0.705^{\pm003}$ | $\underline{0.228}^{\pm023}$ |
| | Ours (A) | $\underline{0.933}^{\pm011}$ | $\mathbf{0.668}^{\pm014}$ | $\mathbf{0.193}^{\pm005}$ | $\mathbf{0.679}^{\pm002}$ | $0.095^{\pm013}$ |
| M-based (U) | (Jang et al., 2022) | $0.658^{\pm009}$ | $0.337^{\pm017}$ | $0.380^{\pm011}$ | $0.857^{\pm004}$ | - |
| | Ours w/o latent | $0.734^{\pm014}$ | $\underline{0.584}^{\pm011}$ | $0.272^{\pm008}$ | $\mathbf{0.721}^{\pm003}$ | - |
| | Ours (V) | $\mathbf{0.860}^{\pm010}$ | $0.499^{\pm015}$ | $\underline{0.221}^{\pm006}$ | $0.747^{\pm004}$ | - |
| | Ours (A) | $\underline{0.814}^{\pm011}$ | $\mathbf{0.588}^{\pm010}$ | $\mathbf{0.217}^{\pm006}$ | $\underline{0.735}^{\pm003}$ | - |
| P-based (U) | Ours w/o latent | - | $\mathbf{0.627}^{\pm014}$ | $0.246^{\pm007}$ | $\underline{0.708}^{\pm003}$ | $\mathbf{1.193}^{\pm029}$ |
| | Ours (V) | - | $0.579^{\pm013}$ | $\underline{0.239}^{\pm006}$ | $\mathbf{0.704}^{\pm002}$ | $0.874^{\pm029}$ |
| | Ours (A) | - | $\underline{0.586}^{\pm015}$ | $\mathbf{0.227}^{\pm006}$ | $0.736^{\pm003}$ | $\underline{0.978}^{\pm026}$ |

Table 3: Quantitative results on the (Xia et al., 2015) test set.

| Setting | Method | Ours wins |
|---|---|---|
| M-based (S) | (Aberman et al., 2020) | 78.69% |
| | (Park et al., 2021) | 73.67% |
| | Ours w/o latent | 65.98% |
| L-based (S) | (Park et al., 2021) | 73.06% |
| M-based (U) | (Jang et al., 2022) | 58.92% |

Table 4: Human evaluation results.

| Methods | Runtime (ms)↓ |
|---|---|
| (Aberman et al., 2020) | 16.763 |
| (Park et al., 2021)(M) | 37.247 |
| (Park et al., 2021)(L) | $\underline{16.329}$ |
| (Jang et al., 2022) | 67.563 |
| Ours (A)(M) | **4.760** |

Table 5: Runtime comparisons.

as content, and randomly sample style motions and labels from (Aberman et al., 2020) test set. For fair comparison, we repeat this experiment 30 times, and report the mean value with a 95% confidence interval. We also consider the variants of our approach: non-latent stylization (*ours w/o latent*), using VAE (*Ours (V)*) or AE (*Ours (A)*) as the latent model (See Sec. 3.1). *Ours w/o latent* employs the identical architecture as our full model, as illustrated in Fig. 2 (a), without the steps of pretraining or training the motion encoder $\mathcal{E}$ and decoder $\mathcal{D}$ as autoencoders. Although it maintains the same number of model parameters, it directly learns style transfer on poses, allowing us to assess the impact of our proposed latent stylization.

Overall, our proposed approach consistently achieves appealing performance on a variety of applications across three datasets. In the supervised setting, GAN approaches, such as (Aberman et al., 2020) and (Park et al., 2021), tend to overfit on one dataset and find it difficult to scale to other motions. For example, (Park et al., 2021) earns the highest achievement on *style recognition* on (Aberman et al., 2020), as 97.1%, while underperforms on the other two unseen datasets, with style accuracy of 81.3% and 79.6%. Furthermore, these methods usually fall short in preserving content, as evidenced by the low content accuracy (31.8% and 44.1%) in Tab. 3. (Jang et al., 2022) is shown to be a strong unsupervised baseline; it gains comparable and robust performance on different datasets, which though still suffers from content preservation. On the contrary, our supervised and unsupervised models commonly maintain high style accuracy over 90% and 80% respectively, with minimal loss on content semantics. Among all variants, *latent* stylization improves the performance on almost all aspects, including generalization ability, with slight compromise on diversity. *Ours (V)* tends to own higher success rate of style transfer, while *ours (A)* typically outperforms on maintaining content (*i.e., Geo Dis* and *Content Accuracy*).

**User Study.** In addition, a user study on Amazon Mechanical Turk is conducted to perceptually evaluate our motion stylization results. 50 comparison pairs (on CMU Mocap (CMU)) between each baseline model and our approach, in the corresponding setting, are generated and shown to 4 users, who are asked to choose their favored one regarding realism and stylization quality. Overall, we collect 992 responses from 27 AMT users who have *master* recognition. As shown in Table 4,

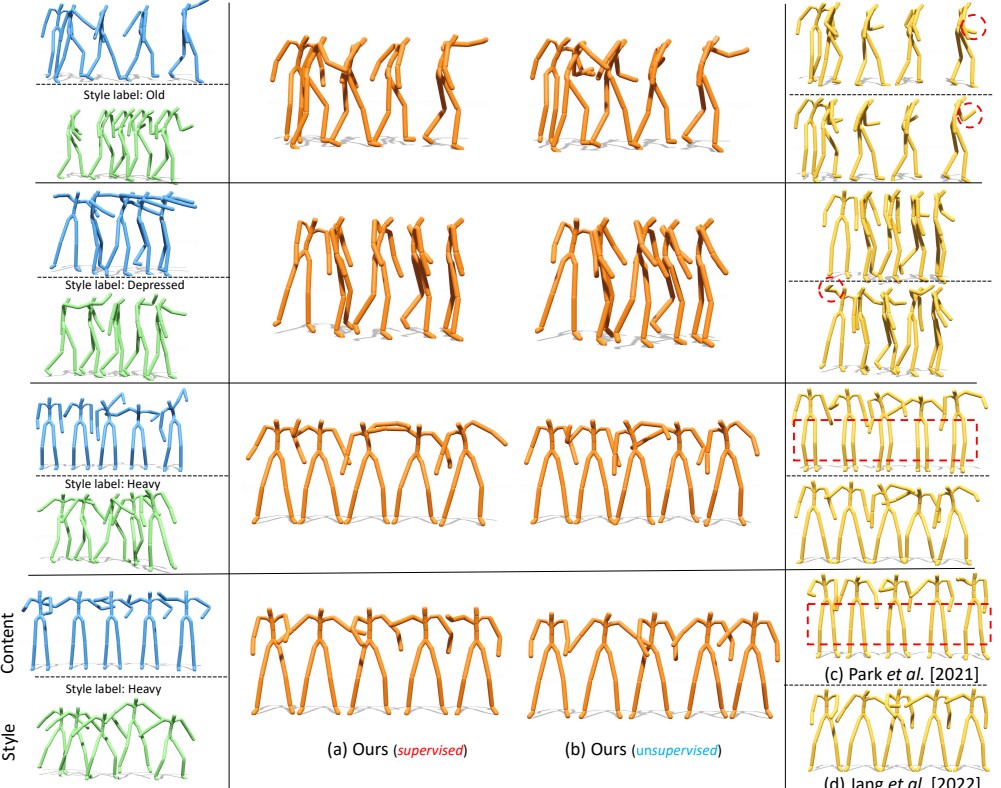

Figure 4: Qualitative comparisons of motion-based stylization. Given the style motion (green) and content motion (blue), we apply stylization using our methods (orange), (Park et al., 2021) (supervised), and (Jang et al., 2022) (unsupervised). The content motions in top two cases come from (Aberman et al., 2020), while the bottom two from CMU Mocap (CMU) test sets. Example artifacts are highlighted using red signs.

our method earns more user appreciation over most of the baselines by a large margin. Further user study details are provided in Appendix F.

**Efficiency.** Table 5 presents the comparisons of average time cost for a single forward pass with 160-frame motion inputs, evaluated on a single Tesla P100 16G GPU. Previous methods apply style injection at each generator layer until the motion output and usually involve computationally intensive operations such as multi-scale skeleton-based GCN and forward-loop kinematics. Benefiting from our latent stylization and lightweight network design, our model appears to be much faster and shows the potential for real-time applications.

## 4.2 QUALITATIVE RESULTS

Figure 4 presents the visual comparison results on test sets of (Aberman et al., 2020) (top two) and CMU Mocap (CMU) (bottom two), in supervised (ours vs. (Park et al., 2021)) and unsupervised (ours vs. (Jang et al., 2022)) settings. For our model, we use *ours(V)* by default. In the unsupervised setting, (Jang et al., 2022) has comparable performance on transferring style from style motion to content motion; but it sometimes changes the actions from content motion, as indicated in red circles. Supervised baseline (Park et al., 2021) follows a similar trend. Moreover, the results of (Park et al., 2021) on CMU Mocap (Aberman et al., 2020) commonly fail to capture the style information from input style motion. This also agrees with the observation of the limited generalization ability of GAN-based models in Tab. 2. Other artifacts such as unnatural poses ( (Aberman et al., 2020; Park et al., 2021)) and foot sliding( (Jang et al., 2022)) can be better viewed in the supplementary video. This can be partially attributed to the baselines directly applying global velocities of content motion for stylization results. In contrast, our approach shows reliable performance in both maintaining content semantics and capturing style characteristics for robust stylization.

**Diverse and Stochastic Stylization.** Our approach allows for diverse label-based and stochastic prior-based stylization. As presented in Figure 5, for label-based stylization, taken one content

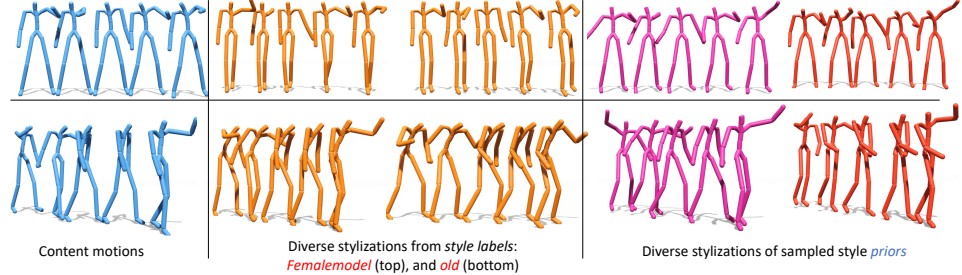

Content motions | Diverse stylizations from *style labels*: *Femalemodel* (top), and *old* (bottom) | Diverse stylizations of sampled style *priors*

Figure 5: Two examples of diverse label-based stylization (middle) and prior-based stylization (right).

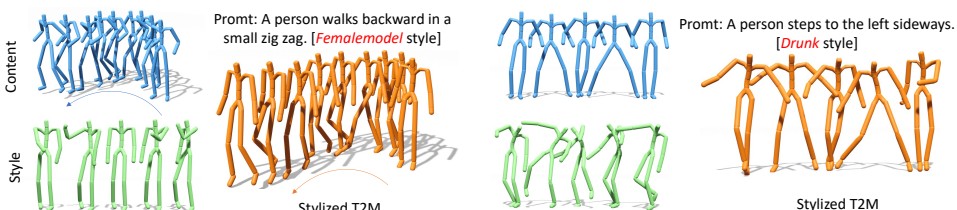

Figure 6: Two stylized text2motion examples, by applying our method behind text2motion (Guo et al., 2022a).

motion and style label as input, our model is able to generate multiple stylized results with inner-class variations, *i.e.,* different manners of *old* man walking. On the other hand, our prior-based stylization can produce results with very distinct styles that are learned unsupervisedly from motion data. These styles are undefined, and possibly non-existent in training data.

**Stylized Text2Motion.** Text-to-motion synthesis has attracted significant interests in recent years (Guo et al., 2022a;b; Petrovich et al., 2022; Tevet et al., 2022); however, the results often exhibit limited style expression. Here in Fig. 6, we demonstrate the feasibility of generating stylistic human motions from the text prompt, by simply plugging our model behind a text2motion generator (Guo et al., 2022a). It is worth noting that the motions from (Guo et al., 2022a) differ greatly from our learning data in terms of motion domain and frame rate (20 fps vs. ours 30 fps).

**Style Code Visualization.** Figure 8 displays the t-SNE 2D projection of our extracted style codes using four model variants, where each sample is color-coded according to its label. In the unsupervised setting, each style code is associated with global style features that are expected to be distinctive with respect to the style category. It can be observed that our *latent stylization* method produces clearer style clusters aligned with style labels compared to our non-latent method, with VAE-based latent model (*ours(V)*) performing the best. While in the supervised setting, as discussed in Sec. 3.2.2, our approach learns label-invariant style features

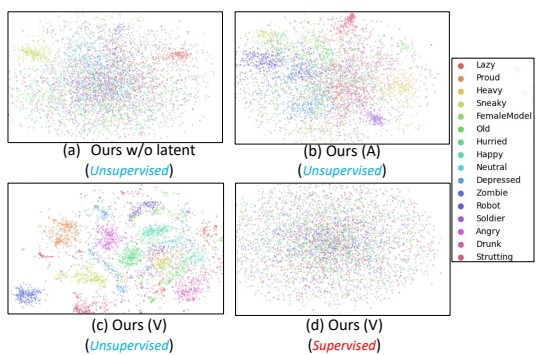

Figure 7: Style code visualization.

(Fig. 8 (d)); These style features may arise from individual and environmental factors.

## 5 CONCLUSION

Our work looks into the problem of 3D human motion stylization, with particular emphasis on generative stylization in the neural latent space. Our approach learns a probabilistic style space from motion latent codes; this space allows style sampling for stylization conditioned on reference style motion, target style label, or free-form novel re-stylization. Experiments on three mocap datasets also demonstrate other merits of our model such as better generalization ability, flexibility in style controls, stylization diversity and efficiency in the forward pass.

**Ethics Statement.** In practice use, our method is likely to cause demographic discrimination, as it involves stereotypical styles related to gender (*Femalemodel*), age (*old*) and occupation (*soldier*).

**Reproducibility Statement.** We have made our best efforts to ensure reproducibility, including but not limited to: 1) detailed description of our implementation details in Appendix D; 2) detailed description of our baseline implementation in Appendix D.3; 3) graphic illustration of our model architecture in Figures 9 and 10; and 4) uploaded codes as supplementary files.

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

This supplementary file provides additional ablation analysis of our design components (Appendix A) and weight of various loss terms (Appendix C), content and feature visualization(Appendix B), implementation details (Appendix D), Evaluation Metric (Appendix E), cross-style and homo-style interpolation results (Appendix G), user study introduction (Appendix F) and failure cases (Appendix H).

**Video.** We also provide several supplementary videos, which contains dynamic animations of our stylization results, visual comparisons, interpolation, stylized text2motion and failure cases. We strongly encourage our audience to watch these videos. It will be much helpful to understand our work. The videos are submitted along with supplementary files, or accessible online (1080P): https://drive.google.com/drive/folders/1UeGuE1qCceLFQJa3vpYoOHC2MoLdBifK?usp=sharing

**Code and Model.** The code of our approach and implemented baselines are also submitted for reference. Code and trained model will be publicly available upon acceptance.

## A    ABLATION ANALYSIS

| S / U | Method | (Aberman et al., 2020) | | | (Xia et al., 2015) | | | |
|---|---|---|---|---|---|---|---|---|
| | | Style Acc↑ | Style FID↓ | Geo Dis↓ | Style Acc↑ | Content Acc↑ | Content FID↓ | Geo Dis↓ |
| S | Ours (A) | $\mathbf{0.945}^{\pm007}$ | $\mathbf{0.020}^{\pm002}$ | $\mathbf{0.344}^{\pm002}$ | $\mathbf{0.926}^{\pm008}$ | $\mathbf{0.674}^{\pm011}$ | $\mathbf{0.189}^{\pm005}$ | $\mathbf{0.680}^{\pm002}$ |
| | *w/o* latent | $\underline{0.932}^{\pm008}$ | $0.022^{\pm002}$ | $0.463^{\pm003}$ | $0.851^{\pm012}$ | $\underline{0.654}^{\pm012}$ | $0.258^{\pm007}$ | $0.707^{\pm003}$ |
| | *w/o* prob-style | $0.913^{\pm007}$ | $0.022^{\pm002}$ | $0.509^{\pm004}$ | $0.870^{\pm010}$ | $0.524^{\pm015}$ | $0.249^{\pm008}$ | $0.767^{\pm004}$ |
| | *w/o* homo-style | $0.883^{\pm012}$ | $0.032^{\pm004}$ | $0.507^{\pm003}$ | $0.851^{\pm012}$ | $0.537^{\pm016}$ | $0.232^{\pm006}$ | $0.760^{\pm004}$ |
| | *w/o* autoencoding | $0.900^{\pm010}$ | $0.026^{\pm002}$ | $0.427^{\pm003}$ | $\underline{0.879}^{\pm010}$ | $0.634^{\pm011}$ | $\underline{0.198}^{\pm005}$ | $0.720^{\pm004}$ |
| | *w/o* cycle-recon | $0.917^{\pm009}$ | $\underline{0.021}^{\pm002}$ | $\underline{0.385}^{\pm003}$ | $0.872^{\pm006}$ | $0.627^{\pm011}$ | $0.208^{\pm004}$ | $\underline{0.699}^{\pm002}$ |
| U | Ours (A) | $\mathbf{0.804}^{\pm011}$ | $\mathbf{0.040}^{\pm003}$ | $\mathbf{0.441}^{\pm003}$ | $\mathbf{0.814}^{\pm011}$ | $\underline{0.588}^{\pm010}$ | $\mathbf{0.217}^{\pm006}$ | $0.735^{\pm003}$ |
| | *w/o* latent | $\underline{0.780}^{\pm014}$ | $0.048^{\pm003}$ | $0.466^{\pm004}$ | $0.734^{\pm014}$ | $0.584^{\pm011}$ | $0.272^{\pm008}$ | $\underline{0.721}^{\pm003}$ |
| | *w/o* prob-style | $0.734^{\pm018}$ | $0.058^{\pm004}$ | $0.461^{\pm003}$ | $0.666^{\pm016}$ | $\mathbf{0.597}^{\pm015}$ | $0.270^{\pm010}$ | $\mathbf{0.718}^{\pm003}$ |
| | *w/o* homo-style | $0.753^{\pm016}$ | $0.050^{\pm002}$ | $0.513^{\pm003}$ | $0.730^{\pm009}$ | $0.526^{\pm013}$ | $0.250^{\pm005}$ | $0.803^{\pm002}$ |
| | *w/o* autoencoding | $0.777^{\pm012}$ | $0.049^{\pm004}$ | $0.493^{\pm004}$ | $\underline{0.811}^{\pm011}$ | $0.491^{\pm015}$ | $\underline{0.230}^{\pm007}$ | $0.759^{\pm005}$ |
| | *w/o* cycle-recon | $0.765^{\pm011}$ | $\underline{0.043}^{\pm004}$ | $\underline{0.560}^{\pm005}$ | $0.756^{\pm017}$ | $0.479^{\pm013}$ | $0.233^{\pm007}$ | $0.869^{\pm002}$ |

Table 6: Ablation study on **different components of our model design**. $\pm$ indicates 95% confidence interval. **Bold** face indicates the best result, while underscore refers to the second best. (S) and (U) denote *supervised* and *unsupervised* setting. Motion-based stylization is presented for both settings. *Prob-style* refers to probabilistic style space.

Table 6 presents the results of ablation experiments investigating various components of our latent stylization models. These components include stylization on the latent space (*latent*), the use of a probabilistic style space (*prob-style*), homo-style alignment (*homo-style*), autoencoding, and cycle reconstruction. The experiments are conducted within the framework of Ours (A) and are focused on the task of motion-based stylization. Results are reported on two datasets (Aberman et al., 2020) and (Xia et al., 2015). It's important to note that the dataset of (Xia et al., 2015) is exclusively used for testing the generalization ability of our models and has not been used during training.

Overall, we observe a notable performance improvement by incorporating different modules into our framework. For instance, our key designs—latent stylization and the use of a probabilistic style space—significantly enhance performance on the unseen (Xia et al., 2015) dataset, resulting in a 7% increase in stylization accuracy in the supervised setting. Additionally, homo-style alignment, despite its simplicity, provides a substantial performance boost across all metrics. Notably, content accuracy sees a remarkable improvement of 13% and 6% in supervised and unsupervised settings, respectively, underscoring the effectiveness of homo-style alignment in preserving semantic information.

In the subsequent sections, we delve into a detailed discussion of three other critical choices in our model architecture and learning scheme: probabilistic (or deterministic) space for content and style features, separate (or end-to-end) training of latent extractor and stylization model, and the incorporation of a global motion predictor.

**Probabilistic Modeling of Style and Content Spaces.** Table 7 presents a comparison between deterministic and probabilistic modeling approaches for both style and content spaces. In our study, the

| Content Space | Style Space | (Aberman et al., 2020) | | | (Xia et al., 2015) | | | |
|---|---|---|---|---|---|---|---|---|
| | | Style Acc↑ | Style FID↓ | Geo Dis↓ | Style Acc↑ | Content Acc↑ | Content FID↓ | Geo Dis↓ |
| D | D | $0.913^{\pm007}$ | $0.022^{\pm002}$ | $0.509^{\pm004}$ | $0.870^{\pm010}$ | $0.524^{\pm015}$ | $0.249^{\pm008}$ | $0.767^{\pm004}$ |
| D | P | $0.945^{\pm007}$ | $0.020^{\pm002}$ | $\mathbf{0.344}^{\pm002}$ | $\mathbf{0.926}^{\pm008}$ | $\mathbf{0.674}^{\pm011}$ | $\mathbf{0.189}^{\pm005}$ | $\mathbf{0.680}^{\pm002}$ |
| P | P | $\mathbf{0.947}^{\pm001}$ | $\mathbf{0.017}^{\pm001}$ | $0.489^{\pm003}$ | $0.891^{\pm003}$ | $0.417^{\pm012}$ | $0.322^{\pm011}$ | $0.758^{\pm003}$ |

Table 7: Ablation study on the **choice of probabilistic (P) or deterministic (D) space** for content and style, in supervised setting. $\pm$ indicates 95% confidence interval. **Bold** face indicates the best result, while underscore refers to the second best. Motion-based stylization is presented.

introduction of a probabilistic style space not only provides remarkable flexibility during inference, enabling diverse stylization and multiple applications, but it also consistently enhances performance and generalization capabilities. An intriguing aspect to explore is the impact of modeling the content space non-deterministically. As highlighted in Tab. 7, we observe that a probabilistic content space achieves superior stylization accuracy on in-domain datasets (Aberman et al., 2020). However, it exhibits sub-optimal generalization performance on out-domain cases (Xia et al., 2015).

| Training Strategy | (Aberman et al., 2020) | | | (Xia et al., 2015) | | | |
|---|---|---|---|---|---|---|---|
| | Style Acc↑ | Style FID↓ | Geo Dis↓ | Style Acc↑ | Content Acc↑ | Content FID↓ | Geo Dis↓ |
| Separately | $\mathbf{0.945}^{\pm007}$ | $\mathbf{0.020}^{\pm002}$ | $\mathbf{0.344}^{\pm002}$ | $\mathbf{0.926}^{\pm008}$ | $\mathbf{0.674}^{\pm011}$ | $\mathbf{0.189}^{\pm005}$ | $\mathbf{0.680}^{\pm002}$ |
| End-to-end | $0.125^{\pm010}$ | $1.521^{\pm024}$ | $0.577^{\pm001}$ | $0.174^{\pm014}$ | $0.293^{\pm002}$ | $1.417^{\pm009}$ | $0.700^{\pm001}$ |

Table 8: Ablation study on **separately or end-to-end training** the latent model and stylization model, in supervised setting. $\pm$ indicates 95% confidence interval. **Bold** face indicates the best result, while underscore refers to the second best. (S) and (U) denote *supervised* and *unsupervised* setting. Motion-based stylization is presented.

**Separate / End-to-end Training.** Our two-stage framework can alternatively be trained in an end-to-end fashion. We also conduct ablation analysis to evaluate the impact of such choice of training strategy. The results are presented in Table 8. In practice, we observed that end-to-end training posed significant challenges. The model struggled to simultaneously learn meaningful latent motion representation and effectively transfer style traits between stages. Experimental results align with this observation, revealing that stylization accuracy is merely around 15% on both datasets in the end-to-end training scenario, in contrast to the accuracy of 92% achieved by stage-by-stage training.

| Method | (Aberman et al., 2020) | | CMU Mocap (CMU) | | (Xia et al., 2015) | |
|---|---|---|---|---|---|---|
| | Style Acc↑ | Foot Skating↓ | Style Acc↑ | Foot Skating↓ | Style Acc↑ | Foot Skating↓ |
| Ours (S) | $\mathbf{0.945}^{\pm007}$ | $\mathbf{0.130}^{\pm001}$ | $0.918^{\pm007}$ | $\mathbf{0.140}^{\pm001}$ | $\mathbf{0.926}^{\pm008}$ | $\mathbf{0.263}^{\pm003}$ |
| Ours *w/o* GMP (S) | $0.942^{\pm003}$ | $0.141^{\pm001}$ | $\mathbf{0.920}^{\pm006}$ | $0.160^{\pm001}$ | $0.882^{\pm008}$ | $0.331^{\pm002}$ |
| Ours (U) | $\mathbf{0.840}^{\pm010}$ | $\mathbf{0.102}^{\pm001}$ | $\mathbf{0.828}^{\pm010}$ | $\mathbf{0.099}^{\pm001}$ | $\mathbf{0.860}^{\pm010}$ | $\mathbf{0.179}^{\pm002}$ |
| Ours *w/o* GMP (U) | $0.817^{\pm013}$ | $0.116^{\pm001}$ | $0.820^{\pm009}$ | $0.122^{\pm001}$ | $0.777^{\pm018}$ | $0.307^{\pm002}$ |

Table 9: Ablation study on **global motion prediction** (GMP, see Sec. 3.2.3). The symbol $\pm$ indicates the 95% confidence interval. **Bold** indicates the best result. (S) and (U) denote *supervised* and *unsupervised* settings, respectively. Results of motion-based stylization are presented. *Foot skating* is measured by the average velocity of foot joints on the XZ-plane during foot contact.

**Global Motion Prediction (GMP).** The primary objective of our global motion prediction is to facilitate adaptive pacing for diverse motion contents and styles. As illustrated in Tab. 10, we quantify the mean square error of GMP in predicting root positions across three test sets, measured in millimeters. Notably, even on the previously unseen dataset Xia et al. (2015), the lightweight GMP performs admirably, with an error of 57.7 mm.

To assess the impact of GMP on stylization performance, we compare against a contrast setting (Ours *w/o* GMP), where global motions are directly obtained from the source content input, akin to previous approaches. Additionally, we introduce a *foot skating* metric to gauge foot sliding artifacts, calculated by the average velocity of foot joints on the XZ-plane during foot contact. Table 9 showcases motion-based results on (Aberman et al., 2020; CMU; Xia et al., 2015) test sets. Across all comparisons, our proposed GMP effectively mitigates foot skating issues. Although 2-dimensional global motion features constitute only a small fraction of the entire 260-dimensional pose vectors,

| (Aberman et al., 2020) | CMU Mocap (CMU) | (Xia et al., 2015) |
|---|---|---|
| 46.2 | 48.7 | 57.7 |

Table 10: **Mean Square Error of Root Position Prediction.** The metric is measured in millimeters. Note the dataset of (Xia et al., 2015) is untouched during the training of the global motion predictor.

it makes considerable difference on the dataset of (Xia et al., 2015), improving the stylization accuracy by around 9%. In our 3rd and 4th supplementary videos, we also illustrate how our GMP enables adaptive pacing in different stylization outcomes (label-based and motion-based) for the same content.

## B  FEATURE VISUALIZATION

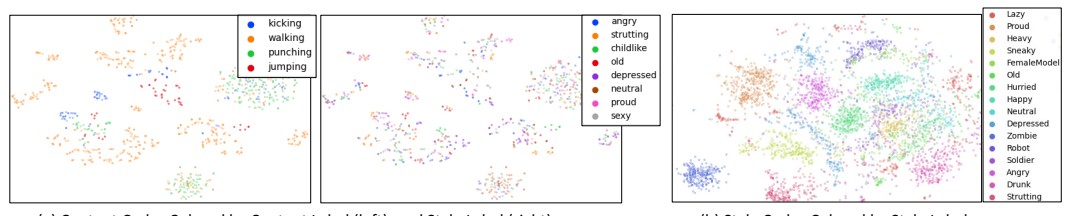

(a) Content Codes Colored by Content Label (left), and Style Label (right).      (b) Style Codes Colored by Style Label.

Figure 8: **Latent Visualization.** Panel (a) displays the projection of the **identical** set of content codes onto a 2D space using t-SNE, colored according to content labels (left) and style labels (right). This visualization suggests that content codes faithfully capture content traits, while style information has been effectively removed. In panel (b), style codes are projected onto a 2D space using t-SNE and colored by their corresponding style labels. Notably, clear style clusters emerge unsupervisedly, aligning with style labels.

Given that our content encoder accommodates motions of arbitrary length, we extract content codes from the Xia et al. (2015) dataset. This dataset, unseen by our models, provides annotations for both style and content labels. Notably, the motions in this dataset are usually short, typically within 3s, which is insufficient to our style encoder. Therefore, for style codes, we takes the motions from dataset (Aberman et al., 2020) for visualization. The models are learned in unsupervised setting, using VAE as latent model.

**Content Code Visualization.** Figure 8 (a) visually presents 2D projections of our content codes. The content codes are colored by their content labels on the left and by their style labels on the right. To generate these projections, the temporal content codes are aggregated along the temporal dimension and then mapped to 2D space using t-SNE. When the content codes are colored by content label (e.g., *walking, kicking*), distinct clusters aligned with the corresponding labels become apparent. However, when the same set of content codes is colored by their style label, these labels are evenly distributed within these clusters. This observation suggests that the content code adeptly captures the characteristics of various contents while effectively erasing style information.

**Style Code Visualization.** Figure 8 (b) visualizes the style codes in a 2D space, color-coded by their style labels. Notably, these style labels were never used during model training. In contrast to the content code visualization in Fig. 8 (a), the projected style codes exhibit a strong connection with the external style label annotations. This observation underscores the effectiveness of our style encoder in extracting style features from the motion corpus.

## C  LOSS WEIGHT ANALYSIS

Tab. 11 presents more quantitative results of our models on (Aberman et al., 2020) and (Xia et al., 2015) test sets. Specifically, we provide the ablation evaluations in both supervised (S) and unsupervised setting (U). For supervised setting, we conduct experiments on label-based stylization which also compares the diversity; and for unsupervised setting we adopt motion-based stylization. Note the base models are not necessarily our final models, here they are set only for reference.

| S / U | $\lambda_{cyc}$ | $\lambda_{kl}$ | $\lambda_{hsa}$ | (Aberman et al., 2020) | | | (Xia et al., 2015) | | |
|---|---|---|---|---|---|---|---|---|---|
| | | | | Style Acc↑ | Geo Dis↓ | Div↑ | Style Acc↑ | Content Acc↑ | Content FID↓ |
| S (base) | 0.1 | 0.01 | 0.1 | $0.937^{\pm008}$ | $0.415^{\pm003}$ | $0.153^{\pm016}$ | $0.913^{\pm008}$ | $\underline{0.669}^{\pm013}$ | $0.202^{\pm006}$ |
| | | | 0.5 | $0.936^{\pm008}$ | $\underline{0.369}^{\pm003}$ | $0.091^{\pm011}$ | $0.924^{\pm007}$ | $\mathbf{0.706}^{\pm010}$ | $\mathbf{0.197}^{\pm007}$ |
| | | 0.001 | | $\mathbf{0.962}^{\pm006}$ | $0.429^{\pm004}$ | $0.125^{\pm016}$ | $\underline{0.933}^{\pm009}$ | $0.619^{\pm014}$ | $\underline{0.197}^{\pm005}$ |
| | | 0.1 | | $0.940^{\pm008}$ | $0.414^{\pm004}$ | $0.141^{\pm015}$ | $0.914^{\pm009}$ | $0.634^{\pm013}$ | $0.209^{\pm005}$ |
| | 0.01 | | | $\underline{0.955}^{\pm006}$ | $0.419^{\pm003}$ | $0.107^{\pm011}$ | $\mathbf{0.957}^{\pm007}$ | $0.609^{\pm011}$ | $0.207^{\pm006}$ |
| | 1 | | | $0.880^{\pm011}$ | $0.423^{\pm003}$ | $\underline{0.302}^{\pm026}$ | $0.833^{\pm011}$ | $0.625^{\pm013}$ | $0.236^{\pm006}$ |
| U (base) | 1 | 0.01 | 0.1 | $\mathbf{0.804}^{\pm011}$ | $0.441^{\pm003}$ | - | $\mathbf{0.814}^{\pm014}$ | $0.588^{\pm010}$ | $0.217^{\pm006}$ |
| | | | 0.01 | $\underline{0.790}^{\pm015}$ | $0.489^{\pm004}$ | - | $0.761^{\pm012}$ | $0.567^{\pm016}$ | $0.224^{\pm007}$ |
| | | 0.1 | | $0.659^{\pm018}$ | $0.430^{\pm004}$ | - | $0.701^{\pm014}$ | $0.619^{\pm013}$ | $\mathbf{0.190}^{\pm005}$ |
| | 0.01 | | | $0.669^{\pm013}$ | $\mathbf{0.388}^{\pm003}$ | - | $0.671^{\pm015}$ | $\mathbf{0.641}^{\pm012}$ | $\underline{0.206}^{\pm006}$ |
| | 0.1 | | | $0.739^{\pm015}$ | $\underline{0.420}^{\pm004}$ | - | $\underline{0.762}^{\pm016}$ | $\underline{0.619}^{\pm014}$ | $0.214^{\pm007}$ |

Table 11: Effect of hyper-parameters of *ours (A)* on the (Aberman et al., 2020) and (Xia et al., 2015) test sets. $\pm$ indicates 95% confidence interval. **Bold** face indicates the best result, while underscore refers to the second best. (S) and (U) denote *supervised* and *unsupervised* setting. For (S), we present results of label-based stylization; and for (U), we present motion-based stylization.

**Effect of $\lambda_{hsa}$.** Homo-style alignment ensures the style space of the sub-clips from one motion sequence to be close to each other; it is an important self-supervised signal in our approach. Increasing the weight of homo-style commonly helps style modeling (style accuracy) and content preservation (content accuracy, FID), which however also comes with lower diversity. A common observation is that the performance on style and content always contradicts with the diversity. It could be possibly attributed to the inherently limited diversity in our training dataset (Aberman et al., 2020), which is collected by one person performing several styles.

**Effect of $\lambda_{kl}$.** $\lambda_{kl}$ weighs how much the overall style space aligns with the prior distribution $\mathcal{N}(\mathbf{0}, \mathbf{I})$. Smaller $\lambda_{kl}$ usually increases the capacity of the model exploiting styles, which on the other hand deteriorate the performance on content maintenance and diversity.

**Effect of $\lambda_{cyc}$.** Cycle reconstruction constraint plays an important role in unsupervised setting. In supervised setting, strong cycle reconstruction constraint is detrimental to style modeling. In contrast, while learning unsupervisedly, strengthening the cycle constraint enhances the performance on style transferring, and at the same time compromises the preservation of content.

| $\lambda_{l1}$ | $\lambda_{sms}$ | (Aberman et al., 2020) | | | (Xia et al., 2015) | | | |
|---|---|---|---|---|---|---|---|---|
| | | MPJPE (Recon)↓ | Style Acc↑ | Style FID↓ | MPJPE (Recon)↓ | Style Acc↑ | Content Acc↑ | Content FID↓ |
| 0.001 | 0.001 | **39.4** | $\mathbf{0.945}^{\pm007}$ | $\mathbf{0.020}^{\pm002}$ | **62.5** | $0.926^{\pm008}$ | $0.674^{\pm011}$ | $\mathbf{0.189}^{\pm005}$ |
| 0.1 | 0.1 | 360.1 | $0.862^{\pm010}$ | $0.041^{\pm004}$ | 431.8 | $0.804^{\pm011}$ | $0.589^{\pm012}$ | $0.276^{\pm007}$ |
| 0.01 | 0.01 | 180.4 | $0.873^{\pm010}$ | $0.041^{\pm004}$ | 250.5 | $0.830^{\pm009}$ | $0.656^{\pm012}$ | $0.244^{\pm007}$ |
| 0.0001 | 0.0001 | 77.6 | $0.857^{\pm010}$ | $0.042^{\pm003}$ | 130.9 | $0.901^{\pm011}$ | $0.661^{\pm013}$ | $0.239^{\pm007}$ |

Table 12: Effect of hyper-parameters of autoencoder on the (Aberman et al., 2020) and (Xia et al., 2015) test sets. $\pm$ indicates 95% confidence interval. **Bold** face indicates the best result, while underscore refers to the second best. Results of motion-based stylization in supervised setting are presented. MPJPE is measured in millimeter.

**Effect of Autoencoder Hyper-Parameters.** In Tab. 12, we investigate the impact of autoencoder hyper-parameters ($\lambda_{l1}$ and $\lambda_{sms}$) on both motion reconstruction and stylization performance. Specifically, $\lambda_{l1}$ encourages sparsity in latent features, while $\lambda_{sms}$ enforces the smoothness of temporal features. Through experimentation, we identify an optimal set of hyper-parameters with $\lambda_{l1} = 0.001$ and $\lambda_{sms} = 0.001$, which yields optimal performance in both reconstruction and stylization tasks. Notably, imposing excessive penalties on smoothness and sparsity proves detrimental to the model's capabilities, resulting in lower reconstruction quality. Additionally, we observe a substantial correlation between reconstruction and stylization performance, indicating that better reconstruction often translates to improved stylization.

# D IMPLEMENTATION DETAILS

Our models are implemented by Pytorch. Motion encoder $\mathcal{E}$ and decoder $\mathcal{D}$ consists of 2 1-D convolution layers; global motion regressor is a 3-layer 1D convolution network. The content encoder $E_c$ and style encoder $E_s$ are also downsampling convolutional networks, where style encoder contains a average pooling layer before the output dense layer. The spatial dimensions of content and style code are both 512. Detailed model architecture is provided in Figs. 9 and 10. The values of $\lambda_{kld}^l$, $\lambda_{l1}$ and $\lambda_{sms}$ are all set to 0.001, and dimension $D_z$ of $\mathbf{z}$ is 512. During training our latent stylization network, the value of $\lambda_{hsa}$, $\lambda_{cyc}$ and $\lambda_{kl}$ are (1, 0.1, 0.1) and (0.1, 1, 0.01) in supervised setting and unsupervised setting, respectively.

## D.1 MODEL STRUCTURE

The detailed architectures of our motion latent auto-encoder and motion latent stylization model are illustrated in Figure 9 and Figure 10 respectively, where "w/o N", "IN" and "AdaIN" refer to without-Normalization, Instance Normalization and Adaptive Instance Normalization operations (Huang & Belongie, 2017). Dropout and Activation layer are omitted for simplicity.

## D.2 DATA PROCESSING

We mostly adopt the pose processing procedure in (Guo et al., 2022a). In short, a single pose is represented by a tuple of root angular velocity, root linear velocity, root height, local joint positions, velocities, 6D rotations (Zhou et al., 2019) and foot contact labels, resulting in 260-D pose representation. Meanwhile, all data is downsampled to 30 FPS, augmented by mirroring, and applied with Z-nomalization.

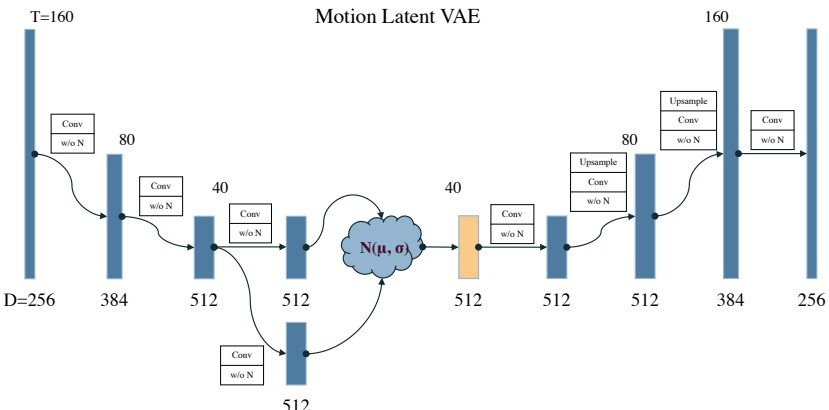

Figure 9: Detailed architecture of our VAE based motion latent model. The AE based latent model keeps only one convolution branch before the latent space. All convolutions, except the last layer of encoder, decoder and generator, use kernel size of 3.

## D.3 BASELINE IMPLEMENTATION

For a fair comparison, we adapt the baseline models with minimal changes from their official implementations, training them on the same data splits. More specifically, without violating their design of input representation and networks, all the re-implemented baseline methods strictly load the same preprocessed data for training.

**(Aberman et al., 2020).** Due to the intentional dual representations for style and content inputs in (Aberman et al., 2020), we make some modifications in the dataloader. We first recover the raw 21-joints structural motion data from the preprocessed data, and convert them into 84-D rotation-based content feature along with 4-D global motion, and 63-D position-based style feature, using their motion parsing function. In addition, we modified the channels of network input/output layers to fit the adapted data. Since our experiments solely consider style from 3-D motions, we disable the 2-D branch as well as the related loss functions. However, we suffer from extremely unstable training process and poor results using the same hyper-parameters. It may result from the length extension

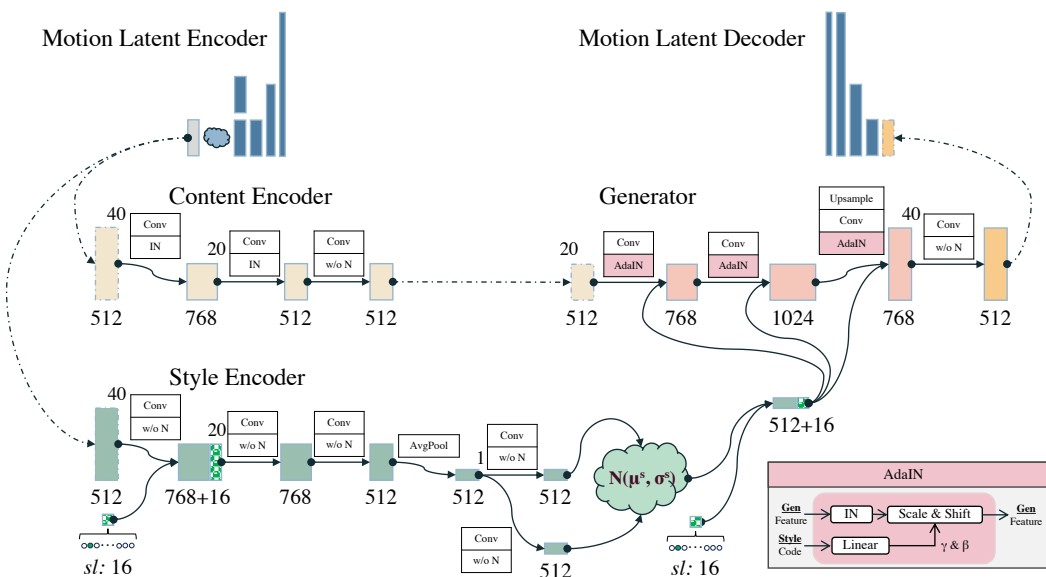

Figure 10: Detailed architecture of our motion latent stylization model in supervised setting. In unsupervised setting, the style label input is dropped. All convolutions, except the last layer of encoders and generator, use kernel size of 3.

of motion sequence (now 160 vs original 32) and inherent flaws of GANs (Zhu et al., 2017; Karras et al., 2019). Thus, empirically, we lower the coefficient for adversarial loss $\alpha_{adv}$ from 1 to 0.5, and update the frequency of discriminator training from 1 per-iteration to 0.2 per-iteration.

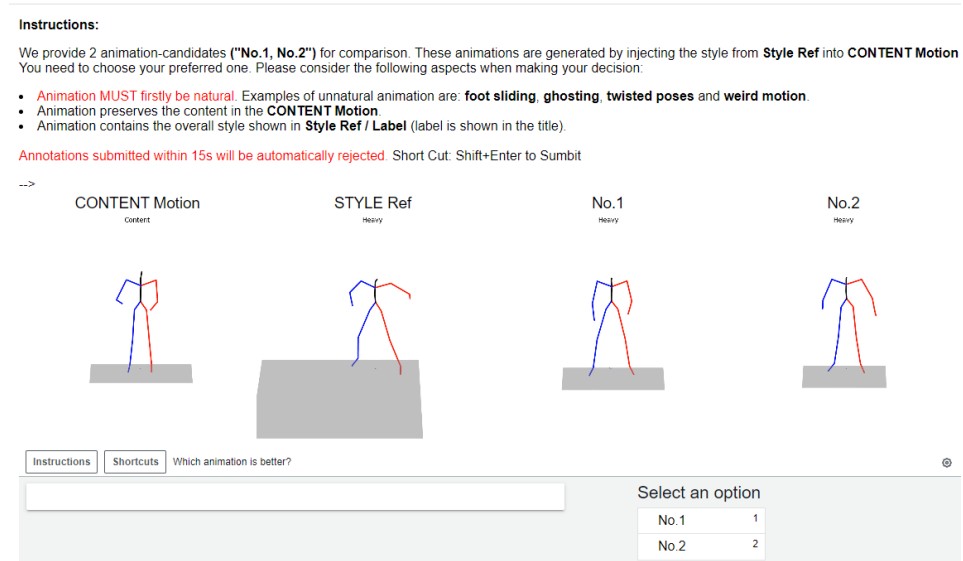

Figure 11: User study interface on Amazon Mechanical Turk.

**(Park et al., 2021).** We extract 63-D joint position feature and 126-D joint rotation feature from our preprocessed data, catering for the designated dataloader in (Park et al., 2021). Meanwhile, we replace the original 4-D quaternion with the equally functional 6-D rotation (Zhou et al., 2019) without any loss of capability. Their model design is limited to fixed motion length, due to the un-scalable linear layer. Therefore, during evaluation on (Xia et al., 2015) test set, we duplicate the sequence to meet the 160-length setting and then extract the corresponding result from the output.

**(Jang et al., 2022)** takes the motion representation building from per-joint's 6-D position-based feature (i.e. position, velocity) and 6-D rotation-based feature (i.e. upward direction, forward direction)

which is almost coherent with our preprocessed data. Thus, we directly re-organize our data to serve the baseline (Jang et al., 2022), keeping everything else unchanged.

## E    EVALUATION METRIC

**Why certain metrics are not used across all datasets?** Given our latent stylization models are trained on  (Aberman et al., 2020), CMU Mocap (CMU) and  (Xia et al., 2015) aims to emphasize zero-shot performance on Style Precision and Content Preservation respectively. Style classifier is trained on  (Aberman et al., 2020), where all style motions come from. Compared to  (Aberman et al., 2020) and CMU Mocap,  (Xia et al., 2015) is quite small (570 clips), comprising variable-length short motions ($< 3s$). Style FID isn't computed for  (Xia et al., 2015) due to substantial length differences between the style motion (from (Aberman et al., 2020), 5.3s) and output motion ($< 3s$). Content classifier is trained only on  (Xia et al., 2015) to evaluate the Content Preservation as content labels are only available on this dataset. Since there is no evidence that this content classifier can generalize to other datasets, we only use it for  (Xia et al., 2015).

## F    USER STUDY

The interface of the user study on Amazon Mechanical Turk for our experiments is shown in Figure 11. Since motion style is not as obvious as other qualitative attributes for common users, to simplify the study, we only compare one baseline result with ours each time. Moreover, for introduction, we briefly explain the concept of motion stylization, presenting the content motion as well as style motion for reference. Users are instructed to choose their preferred results over two generated stylization results based on judgement on *naturalism*, *content preservation* and *style visibility*. This study only involves users that are recognized as **master** by AMT.

## G    INTERPOLATION

We present the results of interpolation in the respective style spaces learned unsupervisedly Fig. 12(a) and supervisedly Fig. 12(b). We are able to interpolate between styles from different labels in unsupervised setting. Specifically, two style codes are extracted from *sneaky* motion and *heavy* motion respectively. Then we mix these two style codes through linear interpolation, and apply them to stylize the given content motion. In supervised setting, the generator is conditioned on a specific style label. Here, we interpolate styles between two random style codes sampled from the prior distribution $\mathcal{N}(\mathbf{0}, \mathbf{I})$. Stylization results are produced conditioned a common style label, *heavy*. From  Figure 12, we can observe the smooth transitions along the interpolation trajectory of two different style codes. Please refer to our supplementary video for better visualization.

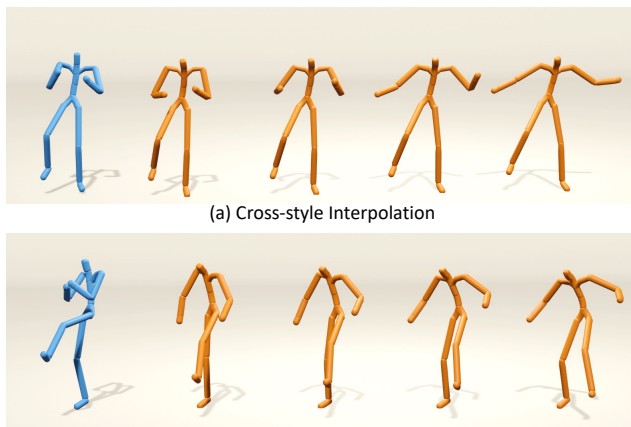

(a) Cross-style Interpolation

(b) Homo-style Interpolation

Figure 12: **Style Interpolation. (a)** Cross-style interpolation in unsupervisedly learned style space. Styles are interpolated between style codes of *sneaky* (left) and *heavy* (right) motions. **(b)** Homo-style interpolation in supervisedly learned style space. With style label *heavy* as condition input, styles are interpolated between two style codes that randomly sampled from $\mathcal{N}(\mathbf{0}, \mathbf{I})$. One key pose for each motion is displayed.

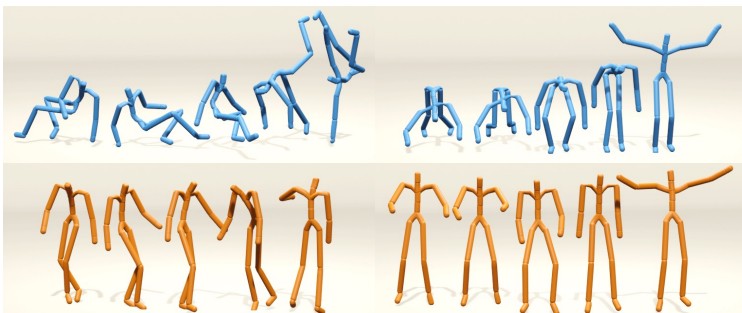

Figure 13: Failure cases. Top row shows content motion; bottom row shows our corresponding results. Stylization results of breaking dance motion (left) and push-up motion (right) using *happy* style label are displayed.

## H    LIMITATIONS AND FAILURE CASES

Firstly, our model may encounter difficulties when the input motion substantially deviates from our training data. Figure 13 presents two failed stylization results on rare content actions, *i.e.,* breaking dance and push-up. Given that our model has only seen standing motions during training, it commonly fails to reserve the lower-body movements in these two cases. Interestingly, our model can still retain the general motions of upper-body.

Secondly, the underlying reason for different performance of ours(V) and ours(A) on for example, diversity, style and content accuracy, remains unclear.

Lastly, certain styles are inherently linked to specific content characteristics, particularly within the datasets of (Aberman et al., 2020; Xia et al., 2015). For instance, styles like old, depressed and lazy typically relate to slow motions, while happy, hurried, angry motions tend to be fast. As our stylization process aims to preserve content information, including speed, there could be contradictions with these style attributes. For instance, stylizing an slow motion with a hurried style might not yield an outcome resembling a hurried motion. We acknowledge this aspect for potential exploration in future studies.

