# OpenReview forum: "Generative Human Motion Stylization in Latent Space"
_ICLR.cc/2024/Conference — ICLR 2024 poster_

### Official Review · Reviewer_fyTQ · 2023-10-24

**Soundness:** 3 good
**Presentation:** 3 good
**Contribution:** 3 good
**Rating:** 5
**Confidence:** 4

**Summary:**

This paper tackles the problem of stylizing 3D human motions. To this end, the authors first train an autoencoder to bring the motion sequence into latent space and propose to decouple the style and content of the motion in latent space. Specifically, the latent code is mapped to content code and style space via two encoders and then mapped back to the original latent space via a generator equiped with adaptive instance normalization (controled by the style code).  During training, several losses are applied to ensure the disentanglement between the content and style including  the Homo-style Alignment loss, swap and cycle reconstruction loss.

**Strengths:**

- The presentation is clear and easy to follow.
- The overall pipeline is sound and the authors provide comprehensive experimental analysis on the performance of the proposed method.

**Weaknesses:**

- In terms of the pipeline itself, there are not much novelties. Every component in the pipeline is straightforward and proposed by previous methods for example the adaptive instance normalization, the cycle consistent reconstuction. Although the proposed homo-style alignment loss is new, it is quite simple.
- More details about the "Ours w/o latent" model is needed.
- It seems all the inline references used in the paper are not in parenthesis. I believe the authors miss used \citet when there should be \citep.

**Questions:**

- It seems the whole model can be trained end-to-end. Why the motion autoencoder is pretrained? What will happend when every components are trained end-to-end?

---

> ### Author Response · Authors · 2023-11-19
> **Reply to Reviewer fyTQ**
>
> **We appreciate your time and effort spent on reviewing our paper. Below, we answer the raised questions one by one.**
>
> ***Q1**: In terms of the pipeline itself, there are not much novelties. Every component in the pipeline is straightforward and proposed by previous methods for example the adaptive instance normalization, the cycle consistent reconstuction. Although the proposed homo-style alignment loss is new, it is quite simple.*
>
> **A1**: We would like hightlight the following technical distinctions of our approach from existing works:
> *  **Latent Stylization**: We propose latent stylization which use the latent features from autoencoders instead of raw poses (as in previous works) for stylization. This compact stylization representation contains more discriminative features, and empirically enhances the performance on stylization accuracy, content preservation, and generalization ability on unseen datasets. It also encourages better style space learning as shown in Figure 7.
> * **Probabilistic Style Encoding**: Unlike previous works using deterministic codes, we propose probabilistic style encoding, modeling style as a Gaussian distribution for each motion input. During training, we encourage all the style Gaussian distribution to be close to N(**0**, **I**). This lead to a smooth and dense style space, supporting novel and diverse style sampling. Benefiting from the probabilistic style, our method exhibits notable flexibility that previous works didn't have, accommodating diverse stylization, various style inputs, and different training settings as shown in Table 1.
>
> * **Homo-style alignment** and **global motion prediction**.  We introduce *homo-style alignment* to provide extra strong supervision signals of style cues which is essentially important in unsupervised setting, and the *post-global motion prediction* to address practical root motion transfer issues.
>
> These technical designs, while not involving computationally intensive operations, achieve state-of-the-art performance while being the most efficient during use. More details are available in our ablation study (Appendix A.), highlighting the performance improvements brought about by these design choices.
>
> ---
>
> ***Q2**:  More details about the "Ours w/o latent" model is needed.*
>
> **A2**: Thanks for the suggestion. We have upload more ablation comparison regarding "ours w/o latent". Please refer to the revised submission of supplementary videos, with file name "7_additional_results(rebuttal).mp4". Or you could easily access through this [anonymous link](https://drive.google.com/file/d/1JcceadXfFwLgDwnlLf0yZ-L_tQ02ZShh/view?usp=drive_link).
>
> ---
> ***Q3**: It seems all the inline references used in the paper are not in parenthesis. I believe the authors miss used \citet when there should be \citep.*
>
> **A3**: We thank the reviewer for pointing this out. We have corrected the reference format in the submitted revision.
>
> ---
> ***Q4**: It seems the whole model can be trained end-to-end. Why the motion autoencoder is pretrained? What will happend when every components are trained end-to-end?*
>
> **A4**: While it is true that the entire model can be trained end-to-end, we found it challenging for the model to simultaneously learn a meaningful latent representation and figure out how to transfer style traits in these latent-based representations. During end-to-end training, we observed the loss quickly getting stuck at a high range. Conversely, training the autoencoders and stylization models stage-by-stage essentially decomposes the whole task into two simpler sub-tasks, allowing each model to address its corresponding aspect. This strategy significantly eases the learning process.
>
> We conducted an ablation experiment to study this choice of training strategy, reporting results on Aberman(*) and Xia(#) datasets in supervised motion-based stylization. The experimental results align with our observation, indicating that stylization accuracy is merely around 15% on both datasets in the end-to-end training scenario, compared to the accuracy of 92% achieved by stage-by-stage training.
>
> | Training Strategy | Style Acc*| Style FID*|  Geo Dist*| Style Acc#| Content Acc# | Content FID#| Geo Dist#|
> | ----  |----  | ----  | ----  | ----  |----  | ----  | ----  |
> | Seperately|**0.945**|**0.020**|**0.344**|**0.926**|**0.674**|**0.189**|**0.680**|
> |End-to-end|0.125|1.521|0.577|0.174|0.293|1.417|0.700|
> ---
>
> This table and related discussion has been incoporated in the revised manuscript, as in Table 8.
>
> ---
> *Thanks again for your value time and effort. Please let us know if you have further concerns.*

---

> ### Comment · Reviewer_fyTQ · 2023-11-21
> **Thank you for the rebuttal**
>
> Although I am still not fully convinced by the novelty of the proposed pipeline, the rebuttal addressed some of my concerns and I now have a clearer view of the paper.  I'd much appreciate it if my further comments can be addressed.
>
> - It would be better to include more technical details of the baseline model "Ours w/o latent" not only the qualitative results. Otherwise, it is hard for readers to understand why it is worse than the proposed model.
>
> - Since the pre-trained auto-encoder cannot be perfect, it is better to report the reconstuction results of the auto-encoder and analyse how the reconstuction quality will affect the style transfer. For example,  what if the model is given an out-of-distribution motion, how will it peform?

---

> ### Author Response · Authors · 2023-11-22
> **Thanks for your follow-up comments**
>
> Dear Reviewer  fyTQ,
>
> **Thanks for your timely response. We hope the following could clarify your concerns.**
>
> ***Q1**: It would be better to include more technical details of the baseline model "Ours w/o latent" not only the qualitative results. Otherwise, it is hard for readers to understand why it is worse than the proposed model.*
>
> **A1**: Thanks for your suggestion on this issue. We agree that this part need further clarification. **Ours w/o latent** employs the identical architecture as our full model, as illustrated in Figure 2 (a), without the steps of pretraining or training the motion encoder $\mathcal{E}$ and decoder $\mathcal{D}$ as autoencoders. Although it maintains the same number of model parameters, it directly learns style transfer on poses, allowing us to assess the impact of our proposed latent stylization.
>
> We also have revised the content in the section 4.1 of updated manuscript.
>
> ---
>
> ***Q2**: Since the pre-trained auto-encoder cannot be perfect, it is better to report the reconstuction results of the auto-encoder and analyse how the reconstuction quality will affect the style transfer. For example, what if the model is given an out-of-distribution motion, how will it peform?*
>
> **A2**: We quantitatively evaluated the reconstruction results of our VAE and AE models using the MPJPE metric in millimeters, obtaining values of 39.4 for AE and 32.5 for VAE, respectively. To analyze the impact of reconstruction quality on stylization, we experimented with different hyperparameters, including $\lambda_{l1}$ and $\lambda_{sms}$ for the AE model. Specifically, $\lambda_{l1}$ encourages sparsity in latent features, while $\lambda_{sms}$ enforces temporal feature smoothness. Results on Xia et al (#) and Aberman et al (*) datasets in the supervised motion-based stylization setting are reported in Table 12 of the revised manuscript.
>
> |$\lambda_{l1}$|$\lambda_{sms}$|MPJPE(*)|Style Acc(*)|Style FID(*)|MPJPE(#)|Style Acc(#)|Content Acc(#)| Content FID(#) |
> |---|---|---|---|---|---|---|---|---|
> |0.001|0.001|**39.4**|**0.945**|**0.020**|**62.5**|**0.926**|**0.674**|**0.189**|
> |0.1|0.1|360.1|0.862|0.041|431.8|0.804|0.589|0.276|
> |0.01|0.01|180.4|*0.873*|*0.041*|250.5|0.830|0.656|0.244|
> |0.0001|0.0001|*77.6*|0.857|0.042|*130.9*|*0.901*|*0.661*|*0.239*|
> ---
> Excessive penalties on smoothness and sparsity proved detrimental, resulting in lower reconstruction quality. We observed a substantial correlation between reconstruction and stylization performance, with better reconstruction often translating to improved stylization, especially on the unseen dataset (i.e, Xia et al.)
> For qualitative results, we introduced a new section 'reconstruction' in the supplementary videos, showcasing reconstructed results from our VAE and AE models.  You may check the newly submitted the supplementary videos, or access this [anonymous link](https://drive.google.com/file/d/1JcceadXfFwLgDwnlLf0yZ-L_tQ02ZShh/view), at the end of the video.
>
> For out-of-distribution motions, we would like to point out that we have provided many such cases in our supplementary videos. Note the motions from CMU Mocap, text2motion results, and the dataset of Xia et al. are all out-of-domain data for our stylization models. To highlight this, in the updated supplemantary videos, we explicitly indicate the motions come from out-of-distributions in the corresponding cases. In additional, for content that are severely different from the training data, such as backflip, breaking dance, the model may fail, as indicated in our discussion of failure cases.
>
> ---
>
> *Thanks again for your comment. We sincerely hope our response have properly addressed your concerns.*

---

> > ### Author Response · Authors · 2023-11-22
> > **Updated video link**
> >
> > Dear Reviewer fyTQ,
> >
> > We found the video link about reconstruction results in our previous response was not updated. We have corrected it. The supplementary videos and manuscript have been updated as well.
> >
> > Meanwhile, the author-reviewer dicussion deadline is approaching. We are eager to know if our response has properly answered your questions. Please feel free to let us know if not.
> >
> > Best,
> >
> > Paper 4166 Authors

---

> > > ### Comment · Reviewer_fyTQ · 2023-11-23
> > >
> > > Thank you for the responses. I have no further questions.

---

### Official Review · Reviewer_bi3M · 2023-10-29

**Soundness:** 3 good
**Presentation:** 3 good
**Contribution:** 2 fair
**Rating:** 6
**Confidence:** 4

**Summary:**

The authors propose a generative method for motion generation. The key idea is using the motion latent code as a compact and expressive representation. The motion code is enforced to decompose into a deterministic content code and a probabilistic style code. The style code is expressed in a certain formula of distribution. And the generation is performed by an auto-encoder. The method can generate stylized motion in different schemes: unconditional or conditional to certain style label or exemplar motion. Also, the benchmarking results also suggest the good effectiveness and the time efficiency of the proposed method.

**Strengths:**

- The proposed method can support two modes of motion generation: conditional to label of style and the by sampling from the prior distribution. This enables a diverse potential downstream applications.
- The latent code of motion is shorter than the motion itself, making the leveraging of motion descriptor in the neural network more efficient.
- The stylization and the content information can be disentangled from the input motion code, supporting versatile application of motion generation.
- I find the idea of using three groups of input is interesting that requires no additional style label but can encourage disentanglement. Also, the cycle-consistency idea is also interesting.

**Weaknesses:**

- Probably not a “weakness”, but this is a unconvincing design to me that a small 1D convolutional network can predict the global motion (say root position) from local joint motion. I would ask for more details and evidence to support this design. The results shown in Table 7 only suggest relatively marginal influence, especially in the supervised settings. Is it possible to also have more results on other datasets?
- Given multiple components and modules are proposed to construct the proposed method, a complete ablation study is expected to validate their effectiveness. The ablation of loss weights shown in Table 6 is probably not enough to convince readers that all proposed components are contributing to the final improvement of the observed benchmarking results.
- Question: The results are focused on motion generation in the skeleton representations. Can the authors also propose some visualization with SMPL LBS? With the mesh, sometimes it can be more intuitive to estimate the plausibility of the joint angles.

**Questions:**

Please see my questions above.

---

> ### Author Response · Authors · 2023-11-19
> **Reply to Reviewer bi3M**
>
> **We thank you for your encouraging comments. The following responses aim to address your concerns point by point.**
>
> ***Q1**: Probably not a “weakness”, but this is a unconvincing design to me that a small 1D convolutional network can predict the global motion (say root position) from local joint motion. I would ask for more details and evidence to support this design. The results shown in Table 7 only suggest relatively marginal influence, especially in the supervised settings. Is it possible to also have more results on other datasets?*
>
> **A1**: **Technical Details**:
> "In this work, our pose representation is a 260-dimensional vector, encompassing rotation, position, and velocity information. The global motion predictor (GMP) focuses on regressing the 2D root velocity on the XY-plane, representing the global motion, based on the remaining 258-dimensional features. All features are Z-normalized. The GMP comprises three 1D convolutional layers with a kernel size of 3, no downsampling, and channel numbers of 512, 256, and 260 at each layer. Training utilizes mean absolute error, with added random Gaussian noise for robustness (scale from 0 to 0.5, probability 0.5).
>
> **More results:** To address your concern, we conducted two experiments to evaluate the GMP. Firstly, we assessed prediction accuracy on three test sets, demonstrating reasonable performance even on unseen datasets (Table 10 in the revised content). We extended the evaluation of GMP to all three datasets (Aberman(*), CMU Mocap(#), and Xia(@)) in motion-based stylization, covering both supervised (S) and unsupervised (U) settings.
>
> |Aberman et al. | CMU Mocap| Xia et al.|
> |---|---|---|
> |46.2|48.7|57.7|
> ---
>
> We further evaluate the combined impact of GMP on the final stylization, and extend the original evaluation to all the three datasets, specifically on Aberman(*), CMU Mocap(#) and Xia(@). This experiment is conducted in motion-based stylization, on both supervised (S) and unsupervised setting (U).
>
> |Method| Style Acc*|Foot Skating*| Style Acc#|Foot Skating#|Style Acc@|Foot Skating@|
> |---|---|---|---|---|---|---|
> |Ours (S)| **0.945** | **0.130**| 0.918| **0.140**|**0.926**|**0.263**|
> |Ours w/o GMP (S)|0.942|0.141|**0.920**|0.160|0.882|0.331|
> |Ours (U)|**0.840**|**0.102**|**0.828**|**0.099**|**0.860**|**0.179**|
> |Ours w/o GMP (U)| 0.817|0.116|0.820|0.122|0.777|0.307|
> ---
>
> We introduced a foot skating metric to assess foot sliding artifacts, observing that our proposed GMP effectively mitigates this issue. While the root motion's impact on style classification results is limited due to its 2D nature in the 260-D pose vector, it improves stylization accuracy on the Xia dataset by 9%. Emphasizing the primary objective of GMP, we illustrate in our 3rd and 4th supplementary videos how it facilitates adaptive pacing for diverse motion contents and styles, showcasing different stylization outcomes (label-based and motion-based) for the same content.
>
> These tables and discussions have been incoporated in the revised manuscript, as in Appendix A., and table 9 and table 10.
>
> ---
> ***Q2**: Given multiple components and modules are proposed to construct the proposed method, a complete ablation study is expected to validate their effectiveness.*
>
> **A2**: We appreciate the reviewer's suggestion and have conducted a comprehensive ablation study on all relevant designs in our model. Below are partial results of motion-based stylization in the supervised setting, and the full comparisons are available in Table 6 of the revised version. Note that symbols * and # represent the Aberman and Xia datasets, respectively.
>
> Methods | Style Acc* |Style FID*|Geo Dist*|Style Acc#|Content Acc#|Content FID#|Geo Dist#
> ---|---|---|---|---|---|---|---
> Ours| **0.945**|**0.020**|**0.344**|**0.926**|**0.674**|**0.189**|**0.680**
> w/o latent|0.932 |0.022|0.463|0.851|0.654|0.258|0.707
> w/o homo-style|0.883|0.032|0.507|0.851|0.537|0.232|0.760
> w/o cycle-recon|0.917|0.021|0.385|0.872|0.627|0.208|0.699
> ---
> ***Q3**: The results are focused on motion generation in the skeleton representations. Can the authors also propose some visualization with SMPL LBS? With the mesh, sometimes it can be more intuitive to estimate the plausibility of the joint angles.*
>
> **A3**: We would like to clarify that SMPL visualization is outside the research scope of this work. For character animation, we have provided visualization examples using clothed 3D avatars in our submitted videos under the name '2_label_based_stylization.' Or you could easily access through this [anonymous link](https://drive.google.com/file/d/1XrpkLLzWHrGHFtJJuLSUELIXBoENdHEM/view?usp=drive_link). Additionally, fitting SMPL shape and poses is not easy in our cases due to the structural differences between our skeleton and SMPL.
>
> ---
> *We sincerely hope that we have properly addressed your concerns. If not, we are happy to open further discussions.*

---

> > ### Comment · Reviewer_bi3M · 2023-11-22
> > **Feedback**
> >
> > Thanks for the reply from the authors to address my concerns.
> >
> > The added experiment results under A1 are great and my concern about the expressiveness of the vector representation has been relieved by the results. I keep my positive rating to this submission.

---

> ### Author Response · Authors · 2023-11-21
> **Additional Response to Reviewer bi3M**
>
> Dear Reviewer bi3M,
>
> Thanks for your time and effort on reviewing our work, and especially your suggestions on additiona analysis of global motion prediction and ablation analysis. The author-reviewer period will be end in two days, we are eager to know if our response adequately addressed your concerns. If you have additional questions, please feel free to let us know.
>
> Best,
>
> Paper 4166 Authors.

---

### Official Review · Reviewer_rCKX · 2023-10-30

**Soundness:** 3 good
**Presentation:** 3 good
**Contribution:** 2 fair
**Rating:** 6
**Confidence:** 3

**Summary:**

The paper presents a new human motion stylization method based on a latent editing approach. The approach is based on a latent motion space which can differentiate the content and style of the motion. Based on the label or prior information of the style, the proposed approach can generate stylized motion results. The proposed approach is validated on several motion benchmarks like Aberman and CMU Mocap. Also, an user study is reported to further validate the performance of the approach.

**Strengths:**

1. The idea of generate stylized motion sequence is interesting and helpful in many industry applications.
2. The proposed approach is well validated in several motion benchmark quantitatively. Also, a user study is provided to further validated the performance of the approach.

**Weaknesses:**

1. How to differentiate between the content and style of motion is not a trivial work. Usually, these two content is interleaved. The proposed approach seems to implicitly to decouple these two factors. Is it possible to provide a validation of the proposed approach can decouple these two content?

2. For the experiments, the metrics may not well validate the performance of the proposed approach. For example, the style accuracy is based on a style classifier, which may be very accurate to capture the styleness of the proposed algorithm. Also, for case with a styled motion input, the transfer result based on the stylized input may not be easily verified.

3. In the attribute edting literature, scholars are actively involved in the style edting, for example, to interpolate two style codes. Is it possible to generalize the generative ability of the proposed approach for more editing operation based on the latent space.

4. How about the performance of the proposed approach on the long motion sequence? Is it possible to maintain the style for the whole motion sequence rather than a short period.

**Questions:**

Please well address the questions in the weakness section.

---

> ### Author Response · Authors · 2023-11-19
> **Reply to Reviewer rCKX (part 1/2)**
>
> **We thank the reviewer for the comments, and hope the following response properly address your concerns.**
>
> ***Q1**: The proposed approach seems to implicitly to decouple these two factors. Is it possible to provide a validation of the proposed approach can decouple these two content?*
>
> **A1**: Explicitly demonstrating the disentanglement of style and content can be challenging due to the absence of a ground truth for these representations. Nevertheless, we recognize the importance of addressing this question and have taken steps to enhance confidence in our model.
>
> To offer insights into the disentanglement, we visualized the extracted content code and style code separately. The reviewer can refer to Figure 8 in the revised manuscript or visit this [anonymous link](https://drive.google.com/file/d/15vnVDOvggH3jyyCVHQRMGO-mA45gA3Ue/view?usp=drive_link). This analysis is conducted using our unsupervised stylization model.
>
>
> For content code visualization, we extracted content codes from the Xia et al. dataset, which includes both content and style labels for each motion sequence. The temporal content codes are summed along the time dimension and projected to 2D space using t-SNE. In subfigure (a), we color the content samples based on their content labels (left) and style labels (right). The left figure demonstrates a strong association between content code clusters and their content labels (e.g., kicking, punching, jumping), while the right figure shows even distribution of style labels within each cluster. This indicates that the content code encapsulates semantic information while removing style identities.
>
> For style code visualization, we extracted style codes from the Aberman dataset, which includes style label annotations. The style codes are similarly projected to 2D space using t-SNE and colored by their corresponding style labels. Notably, these style code clusters align with their labels, even though they are learned unsupervisedly and implicitly. This demonstrates the capability of our style encoder captures the style characteristics from the motion corpus. Unfortunately, the Xia dataset is not suitable for this case due to the shorter durations of its motions compared to the 6s input required for our style encoder.
>
> These analysis and content have been added in our revised manuscript.
>
> ---
>
> ***Q2**: For the experiments, the metrics may not well validate the performance of the proposed approach. For example, the style accuracy is based on a style classifier, which may be very accurate to capture the styleness of the proposed algorithm.*
>
> **A2**: We understand the concern about using a style classifier for stylization accuracy. However, it's important to note that in real-world scenarios, the perception of style is subjective and may not have a universally accepted defintiion. Our use of a style classifier is a practical attempt to quantify and measure style in a consistent manner, even if it may not capture all nuances. It is also widely adopted in the prior works [1, 2, 3, 4, 5], even when dealing with inputting style motions[1,2, 4,5].
>
> Moreover, we acknowledge that relying on a single style metric may have limitations. We have made our best efforts to ensure a fair and comprehensive evaluation. This includes extending existing metrics such as content accuracy, content/style FID, geodesic distance and inference cost. Additionally, we conducted subjective user studies to gather qualitative feedback. We are also open to other ideas if the reviewer think it necessary.
>
> ---
>
> ***Q3**: In the attribute edting literature, scholars are actively involved in the style edting, for example, to interpolate two style codes. Is it possible to generalize the generative ability of the proposed approach for more editing operation based on the latent space.*
>
> **A3**: We would like to remind the reviewer that we have included visualizations of both cross-style and homo-style interpolations in our supplementary videos, specifically in the file named "5_interpolation.mp4." Or you could visit through this [anonymous link](https://drive.google.com/file/d/1ygkcFROeHOb2R8Sog34IAYMMQqnsCk-_/view?usp=drive_link
> ). These visualizations demonstrate the smoothness and compactness of the learned style space.
>
> ---
>
> [1] Unpaired Motion Style Transfer from Video to Animation, ACM Transactions on Graphics (TOG), 2020.
> [2] Motion Puzzle: Arbitrary Motion Style Transfer by Body Part, ACM Transactions on Graphics (TOG), 2022.
> [3] Style-ERD: Responsive and Coherent Online Motion Style Transfer, CVPR, 2022.
> [4] Autoregressive Stylized Motion Synthesis with Generative Flow, CVPR 2021.
> [5] Diverse Motion Stylization for Multiple Style Domains via Spatial-Temporal Graph-Based Generative Model, SCA, 2021.

---

> > ### Comment · Reviewer_rCKX · 2023-11-22
> >
> > Thanks for the responses. The authors have well addressed my concerns in last round. I will shift my rating to the positive side.

---

> ### Author Response · Authors · 2023-11-19
> **Reply to Reviewer rCKX (part 2/2)**
>
> ***Q4**: How about the performance of the proposed approach on the long motion sequence? Is it possible to maintain the style for the whole motion sequence rather than a short period.*
>
> **A4**: Our model is designed to stylize motions with arbitrary length, as the content encoder utilizes a 1D CNN window sliding along the temporal dimension. In our submitted videos, you can find many motions of variable length, such as stylized text2motion. Additionally, we have included two animations specifically showcasing long motion stylization in the revised supplementary videos, labeled as "7_additional_results(rebuttal).mp4." Or you could visit through this [anonymous link](https://drive.google.com/file/d/1JcceadXfFwLgDwnlLf0yZ-L_tQ02ZShh/view?usp=drive_link).
>
> ---
> *Hope our reply satisfactorily address your concerns. Otherwise, we will be happy to further discuss.*

---

> ### Author Response · Authors · 2023-11-21
> **Additional Response to Reviewer rCKX**
>
> Dear Reviewer rCKX,
>
> We sincerely thank you for reviewing our work. This is a kind reminder that the author-reviewer period will be over in two days. Please feel free to let us know if you have any other questions.
>
> Best,
>
> Paper 4166 Authors.

---

### Official Review · Reviewer_rvJ1 · 2023-11-01

**Soundness:** 3 good
**Presentation:** 3 good
**Contribution:** 3 good
**Rating:** 6
**Confidence:** 4

**Summary:**

This paper introduces an innovative framework for human motion stylization. The method breaks down the motion latent code into a content code and a style code. It leverages AdaIN to infuse style information into the semantic content, dictated by the content code. Furthermore, the authors introduce a new learning approach that encompasses reconstruction, cycle consistency, and homostyle alignment.

**Strengths:**

1. The paper demonstrates impressive quantitative and qualitative results, establishing a new state-of-the-art with a substantial margin.

2. The proposed framework is novel. The associated conclusions and experiments make significant contributions to the research community.

3. The paper is well-written, ensuring that its content is easily understandable for readers.

**Weaknesses:**

My main concern primarily revolves around the selection between probabilistic and deterministic methods when extracting the content code and style code. The authors should incorporate further discussion and analysis in the following areas:

1. The authors should provide more intuitive explanations for why deterministic encoding is used for the content code while probabilistic encoding is used for the style code.

2. The authors need to conduct an ablation study to validate their conclusions regarding this design aspect.

**Questions:**

Please kindly refer to the weaknesses mentioned above.

---

> ### Author Response · Authors · 2023-11-19
> **Reply to Reviewer rvJ1**
>
> **Thank you for your encouraging comments and constructive suggestions. They are indeed important to improve the clarity of our work. Meanwhile, your questions are answered as below.**
>
> ***Q1**: The authors should provide more intuitive explanations for why deterministic encoding is used for the content code while probabilistic encoding is used for the style code.*
>
> **A1**: The main reason of using probabilistic encodeing for style code is to deriving generative ability for diverse or novel style sampling. While for content code, we instead use deterministic encoding for two reasons. Firstly, We do not expect generative ability from temporal content space (it is also difficulty to have). Secondly, unlike styles, which often represent global features of motions, content features exhibit more locality and determinism. These features correlate with precise details in local contexts, making deterministic encoding more appropriate and alleviating the learning burden.
>
> We agree this could be important as it's part of our model design. We have added the content in sec 3.2.1.
>
> ---
> ***Q2**:  The authors need to conduct an ablation study to validate their conclusions regarding this design aspect.*
>
> **A1**: Thanks for the sugguestion. For the analysis of this selection, we conduct an ablation experiment on the supervised motion-based stylization setting.  We report the performance on Aberman et al(*) dataset and Xia et al (#) dataset. Note Xia dataset is unseen to model, aiming to test the generalization ability of models. We denote probabilistic encoder as P, and deterministic encoding as D. Firstly, we can see that probabilistic encoding of styles (second row) outperforms the deterministic encoding (first row) on both dataset with significant margins. For instance, the content accuracy achieved by the probabilistic style encoding on the Xia et al. dataset is 15% higher than that of the deterministic counterpart. Expanding our analysis to the content space, modeling it in a similar probabilistic manner (third row) yields nuanced performance gains in stylization accuracy. However, we observed a challenge in generalization to the Xia et al. dataset, resulting in a sharp performance drop. Meanwhile, the performance of probabilistic content encoding on *content presevation* (as indicated in Geo Dist, Content Acc, etc.) is also suboptimal, which also agrees with the assumption we made above. We also have added this table and related dicussion on the result in Table 7 in the Appendix. A.
>
> | Content Space   | Style Space | Style Acc*| Style FID*|  Geo Dist*| Style Acc#| Content Acc# | Content FID#| Geo Dist#|
> | ----  | ----  |----  | ----  | ----  | ----  |----  | ----  | ----  |
> | D | D|0.913|0.022|0.509|0.870|0.524|0.249|0.767|
> |D|P|0.945|0.020|**0.344**|**0.926**|**0.674**|**0.189**|**0.680**|
> |P|P|**0.947**|**0.017**|0.489|0.891|0.417|0.322|0.758|
>
> ---
>
> **We sincerely hope our response addresses all your concerns. Please feel free to let us know if you have further questions.**

---

> > ### Author Response · Authors · 2023-11-21
> > **Additional Response to Reviewer rvJ1**
> >
> > Dear Reviewer rvJ1,
> >
> > We sincerely thank you again for you effort in reviewing our paper, especially for the helpful suggestions on **intuition / abalation** of the choice of probabilistic encoding. We believe this further strengthen our paper. The author-reviewer discussion will be closed in two days. Please do not hesitate to let us know if you have additional questions, we would be more than happy to help on them.
> >
> > Best,
> >
> > Paper 4166 Authors.

---

### Author Response · Authors · 2023-11-19
**General Response to All Reviewers of Paper 4166**

We appreciate the constructive feedback from all the reviewers. Meanwhile, we are encouraged by the positive feedback from reviewers such as **novel framework and interesting designs** (Reviewer rvJ1, bi3M), **sound pipeline** (Reviewer fyTQ), **approach is well validated**(Reviewer rCKX, fyTQ), **presentation is easy-to-follow**(Reviewer rvJ1, fyTQ). In our individual responses, we attempt to address specific questions and comments as clearly and in detail as possible. Here, we provide a general response summarizing the changes we made to improve the quality of our submission and address the reviewers' concerns during the rebuttal.

We would like to re-emphasize the novelty and contributions of this work:

* To the best of our knowledge, we are among the earliest to propose *generative motion stylization*, enabling non-deterministic stylization under both supervised and unsupervised settings. This approach offers notable flexibility, accommodating stylization from style motion, style label or even without explicit style input, all within a single framework. Furthermore, it is light in design and efficient in use, being nearly 14 times faster than the most advanced prior work [1] (4.7ms vs 67.5ms).
* Our introduction of latent stylization, utilizing motion latent features as an intermediate representation for stylization, is novel. It effectively enhances the expressivity and disentanglement of learned features (Figure. 7), and model generalizability (Table 1,2), potentially shedding light on other neural stylization research.
* We propose several strategies, including homo-style alignment and global motion predictor, to facilitate natural and diverse human motion stylization. These techniques address practical challenges in motion tasks. Their effectiveness is also validated through ablation analysis.
* All these together creates a strong framework that achieves state-of-the-art performance on various datasets, setting and applications. Our framework can generalize to unseen datasets (e.g., the dataset of Xia et al.), and real applications (e.g., stylized text2motion).
---
We also take seriously into consideration the comments and suggestions from reviewers and have incorporated the following changes in our revised manuscript. All the modification areas are highlighted using blue colour in the revised version.

1.   Add an intuitive explanation of using probabilistic style space and deterministic content space in Sec 3.2.1, as suggested by reviewer **rvJ1**.
2.  Add the ablation experiment of the selection of probabilistic/deterministic spaces in Table 7, as suggested by reviewer **rvJ1**.
3.  Add the results of the ablation experiment on all relevant designs of our framework in Table 6, as suggested by reviewer **bi3M**.
4.  Measuring the root position prediction error of the global motion predictor in Table 10, for answering reviewer **bi3M**'s question.
5.  Ablation experiment of global motion prediction on all datasets in Table 9, for answering reviewer **bi3M**'s question.
6.  Visualization of content and style features in latent space for a better sense of disentanglement in Figure 8, for answering reviewer **rCKX**'s question.
7.  Qualitative results of long-motion stylization in supplementary videos, for answering reviewer **rCKX**'s question.
8.  Qualitative comparisons between ours/ ours w/o latent in supplementary videos, as suggested by reviewer **fyTQ**.
9.  Ablation experiment on separate/end-to-end training strategy in Table 8, as sugguested by reviewer **fyTQ**.

[1] Motion puzzle: Arbitrary motion style transfer by body part. ACM Transactions on Graphics (TOG), 2022.

We hope that these additional results further strengthen our approach as state-of-the-art motion stylization models and demonstrate its flexibility and generalization ability.

---

### Author Response · Authors · 2023-11-22
**General Response (Follow-up)**

Dear Reviewers,

First of all, we sincerely thank you all for your effort and time on our work. We believe that your comments and suggestions indeed effectively improve the clarity and validity of our methods and experiments. The author-reviewer discussion will end in 24 hours. Please do not hesitate to let us know any of your additional concerns. *If our responses have satisfactorily addressed your concerns, we would appreciate it so much if you could re-evaluate our work, and possibly raise your score.*

Thanks again for your engagement in evaluating our work.

Best,

Paper 4166 Authors.

---

### Meta-Review · Area_Chair_67T5 · 2023-12-25

**Metareview:**

This paper proposes a method for content and style disentanglement for motion. The method uses an autoencoder with swapping to create a latent space with content and style codes. The method allows for random stylization or a given motion, or style transfer.

Initial reviews were mixed, with reviewers split. The reviewers found the overall method to be effective and well-validated. There were concerns regarding architectural decisions of which codes are stochastic or deterministic (rvJ1), how content/style is defined (rCKX) lack of an ablation study (bi3M), and novelty of the method (fyTQ). The authors submitted a rebuttal to each reviewer, and reviewers rCKX and bi3M found that their comments were adequately addressed, with rCKX raising their rating.

After discussion, 3/4 reviewers recommend acceptance. The AC agrees the proposed architecture is novel in the area of human stylization and recommends acceptance.

**Justification For Why Not Higher Score:**

While the architectural contribution itself is a solid contribution to ICLR for motion modeling, it is not novel in the context of overall generative modeling, as style and content disentanglement has been thoroughly explored in the image generation space, for example using swapping (Park, NeurIPS 2020) and cycle consistency (Zhu, NIPS 2017).

**Justification For Why Not Lower Score:**

The majority of reviewers agree that the paper is a solid contribution for the conference.

---

### Decision · Program_Chairs · 2024-01-16

Accept (poster)